# In situ imaging of the atomic phase transition dynamics in metal halide perovskites

Mengmeng Ma [1,2], Xuliang Zhang[1,2], Xiao Chen [3], Hao Xiong [3], Liang Xu[1,2], Tao Cheng [1,2], Jianyu Yuan [1,2], Fei Wei [3] & Boyuan Shen [1,2] ✉

Phase transition dynamics are an important concern in the wide applications of metal halide perovskites, which fundamentally determine the optoelectronic properties and stabilities of perovskite materials and devices. However, a more in-depth understanding of such a phase transition process with real atomic resolution is still limited by the immature low-dose electron microscopy and in situ imaging studies to date. Here, we apply an emergent low-dose imaging technique to identify different phase structures (α, β and γ) in CsPbI₃ nano-crystals during an in-situ heating process. The rotation angles of PbI₆ octa-hedrons can be measured in these images to quantitatively describe the thermal-induced phase distribution and phase transition. Then, the dynamics of such a phase transition are studied at a macro time scale by continuously imaging the phase distribution in a single nanocrystal. The structural evolution process of CsPbI₃ nanocrystals at the particle level, including the changes in morphology and composition, is also visualized with increasing temperature. These results provide atomic insights into the transition dynamics of per-ovskite phases, indicating a long-time transition process with obvious inter-mediate states and spatial distribution that should be generally considered in the further study of structure-property relations and device performance.

The renaissance of metal halide perovskites (MHPs) has lightened new hopes for wide applications in photovoltaics, light-emitting diodes, photodetectors, and lasers[1–7]. The atomic structures in MHPs, which are defined as specific phases with different crystal systems, essentially determine their optoelectronic and semiconductor properties for these applications[8–14]. Controlling the phase transition and phase stability in MHPs have become an important strategy to improve the material properties and device performances[15–18]. The transition between the α, β, and γ phases of MHPs induced by tem-perature is a core issue in material processing, testing and degrada-tion, which has been mainly studied by diffraction methods[19–22], especially synchrotron X-ray diffraction (XRD)[23–26]. Many different factors in the real world can influence the structural changes in

perovskites[9,27–29]. Among them, the thermal-induced phase transition and structural degradation should be the most direct influencing factors in device applications. However, these previous studies by the diffraction methods mainly provided the averaged structural information of macro samples without local structural information, limiting our understanding of atomic phase distribution and phase transition dynamics in MHPs, especially when the non-perovskite phase can act as an impurity to interfere with our positioning of diffraction peaks in perovskite phase.

Real-space imaging methods, such as electron microscopy, can provide a real-time and in situ study of the thermal-induced phase transition with spatial distribution and dynamics. Although aberration-corrected electron microscopy has been proven to achieve atomic-

[1]Institute of Functional Nano & Soft Materials (FUNSOM), Soochow University, 215123 Suzhou, Jiangsu, PR China. [2]Jiangsu Key Laboratory of Advanced Negative Carbon Technologies, Soochow University, 215123 Suzhou, Jiangsu, PR China. [3]Beijing Key Laboratory of Green Chemical Reaction Engineering and Technology, Department of Chemical Engineering, Tsinghua University, 100084 Beijing, PR China. ✉e-mail: byshen@suda.edu.cn

resolution imaging of crystalline structures, there are only a few reports of imaging results[30–38] and no in situ study of MHP structures with real atomic resolution by electron microscopy. On the one hand, MHPs, even inorganic ones, are very sensitive to the electron beam, which requires the balance of the electron dose and signal-to-noise ratio in images. On the other hand, some operations in in-situ experiments, such as calibration and alignment of samples, are still challenging and require personal experience. Recently, we used the integrated differential phase contrast (iDPC) scanning transmission electron microscopy (STEM) for beam-sensitive materials[39,40], and then, combined it with an in situ imaging technique to realize the in situ observation of the adsorption/desorption behaviors of small molecules[41–44]. These advances promoted the applications of these imaging techniques for beam-sensitive materials and laid a foundation for further studies on more structural evolution in these materials, including the MHP phase transition that we are interested in here.

In this work, we use the iDPC-STEM to identify the phase structures of CsPbI$_3$ nanocrystals with atomic resolution. The rotation of the PbI$_6$ octahedron, which is the structural basis of the phase transition, is quantitatively studied by the profile analysis of iDPC-STEM images. During the in situ heating process, the temperature-dependent MHP phase structures are atomically resolved and the rotation angles of PbI$_6$ octahedrons at different temperatures are recorded from two different projections. More importantly, the distribution of phases and their evolution with temperature (also expressed by the rotation of the PbI$_6$ octahedrons) are revealed by continuously imaging a single nanocrystal to investigate the dynamics of the phase transition in CsPbI$_3$ occurring from its surface to the center. Then, the thermal-induced degradation and conversion of MHPs at the particle level are also observed. These results not only provide imaging evidence for the atomic local phase structures in MHPs, but also reveal the space and time scales of MHP phase transition dynamics with a superb resolution higher than expected.

## Results and discussion
### Phase structures of CsPbI$_3$ nanocrystals
First, to study the phase transition in MHPs, the α, β, and γ phases of MHPs should be accurately identified, which need a clear atomic arrangement in at least two different orthogonal projections. The iDPC-STEM exhibits a unique advantage for phase identification compared to traditional modes. When comparing the HAADF- and iDPC-STEM images collected simultaneously with the same low dose (about 1266 e$^-$/Å$^2$) in Supplementary Fig. 1, the light elements (I and Cs) and surface structures can be observed more clearly together with Pb in the iDPC-STEM image due to the imaging contrast linear to the atomic number during the integration process from four original images (Supplementary Fig. 2), which helps us to identify the ion position and other detailed structural information of each phase and local structure. Based on the definition of a-, b-, c-axis in different phases (Supplementary Fig. 3), Fig. 1a shows the structural models and corresponding simulated iDPC-STEM images of these phases of CsPbI$_3$ viewed from different directions. The structural models were obtained from ref. 19 using the synchrotron XRD method. Based on the reference systems given in Fig. 1a, it should be noticed that the defined zone axes of the α phase is different from those of the β and γ phase due to their different crystal systems[45,46]. The α phase of CsPbI$_3$ exhibits a cubic crystalline structure. We only present the results of its [001] projection because the atomic arrangement in the three orthogonal projections of the α phase is the same. For the β and γ phases (tetragonal and orthorhombic, respectively), the iDPC-STEM images viewed from the perpendicular [001] and [110] directions should be different. Combining the images from these two directions, we defined two angles θ$_1$ ([001]) and θ$_2$ ([110]) as shown in Fig. 1a to characterize the rotation of the PbI$_6$ octahedron in three-dimensional space, since such rotation is just the structural origin of the phase transition. In the α phase, θ$_1$ and θ$_2$ should be zero, while they will increase successively when turning into the β and γ phases, respectively.

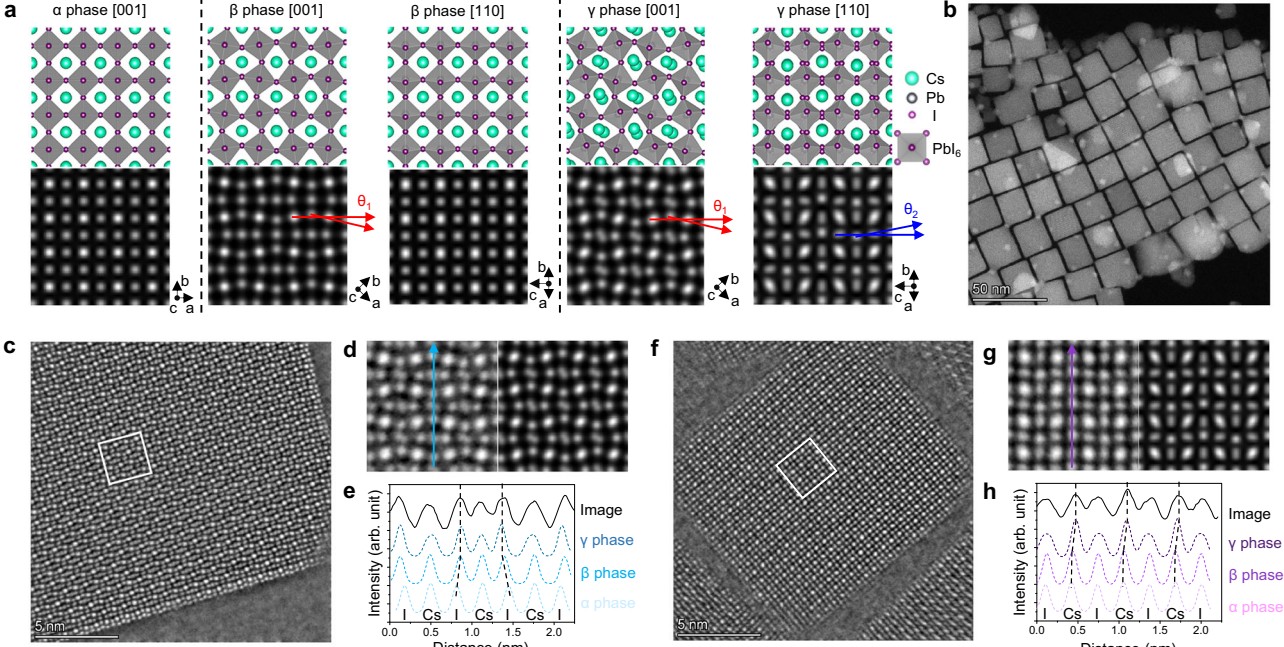

**Fig. 1 | Identifying the phase structures of CsPbI$_3$ nanocrystals. a** Structural models and corresponding simulated iDPC-STEM images of α, β, and γ CsPbI$_3$ phases viewed from different directions. Two angles, θ$_1$ and θ$_2$, are defined in the [001] and [110] projections to reflect the rotation of the PbI$_6$ octahedron, respectively. **b** High angle annular dark field (HAADF) STEM image showing cubic nanocrystals. **c** IDPC-STEM image of CsPbI$_3$ nanocrystals viewed from the [001] direction. **d** Magnified image compared with the simulated image of γ CsPbI$_3$ viewed from the [001] direction. **e** Intensity profile (solid line) extracted along the cyan arrow in (**d**) and compared with the simulated profiles of α, β, and γ CsPbI$_3$ (dashed lines). **f** IDPC-STEM image of CsPbI$_3$ nanocrystals viewed from the [110] direction. **g** Magnified image compared with the simulated image of γ CsPbI$_3$ viewed from the [110] direction. **h** Intensity profile (solid line) extracted along the purple arrow in (**g**) and compared with the simulated profiles of α, β, and γ CsPbI$_3$ (dashed lines). Source data are provided as a Source Data file.

The samples we studied in this work are $CsPbI_3$ nanocrystals with regular cubic shapes and 10–20 nm sizes, as shown in Fig. 1b. Two orthogonal surfaces of the $CsPbI_3$ cube correspond to the [001] and [110] directions, and the thickness is very suitable for STEM characterization. Based on the imaging results in Fig. 1c–h, the $CsPbI_3$ nanocrystals are confirmed to be orthorhombic γ phase at room temperature (R.T.). Figure 1c shows the iDPC-STEM image of a $CsPbI_3$ cube viewed from the [001] direction. Figure 1d shows the magnified image (left) compared with the simulated image (right). Two features can be visually extracted from Fig. 1d to support the γ phase of $CsPbI_3$. One is that the $PbI_6$ octahedrons were obviously rotated to cause nonzero $\theta_1$ as we have defined. Another is that the Cs ion columns were misaligned and imaged as elliptical spots rather than round spots in the α and β phases (compared in Fig. 1a). Meanwhile, the rotation of $PbI_6$ octahedrons can also be expressed by intensity profile analysis in Fig. 1e, which indicates the relative positions of different ion columns. The black solid line shows the profile extracted along the cyan arrow in Fig. 1d, while the colored dashed lines show the simulated results in the α, β, and γ phases along the same direction. Comparing the I peak positions in the experimental and simulated results, there should be in-plane rotation of $PbI_6$ octahedrons in the β and γ phases to make the I peaks on both sides close to the middle Cs peak.

Then, from the imaging results in another [110] projection (Fig. 1f–h), we find that the Cs columns are slightly offset from the center of four I columns (so that they cause nonzero $\theta_2$ as we have defined) by comparing them with the simulated image of the γ phase (Fig. 1g), which is just due to the rotation of $PbI_6$ octahedrons out of the (001) plane (a characteristic of the γ phase). This is also expressed in the intensity profiles in Fig. 1h, where only the simulated profiles of the γ phase show the same Cs peak offset as the experimental profile. Thus, we can finally determine that this $CsPbI_3$ sample is the orthorhombic γ phase at R.T. by combining the images of two projections. In the following studies on the phase transition, we also tried to obtain comprehensive structural information from these two projections. For example, the change in $\theta_1$ reflects the transition from the β to α phase, while the change in $\theta_2$ reflects that from the γ to β phase, which directly connects the rotation angle of local $PbI_6$ octahedrons with the change in phase and its distribution. Such method was also used previously to identify perovskite phases by their structural features[32,37].

## Thermal-induced perovskite phase transition

After putting the samples in an in situ heating chip, we found that from R.T. to 200 °C, the sample will maintain the perovskite structure of $CsPbI_3$ and transform among these three phases. After heating to 250 °C, the morphology of the nanocrystals will undergo drastic changes and finally be converted into other substances. Therefore, we studied the phase transition of $CsPbI_3$ in a temperature range of R.T. to 250 °C by the iDPC-STEM and an in situ heating program. Figure 2a–c shows the atomic arrangement in the [001] projection of $CsPbI_3$ at R.T., 100 and 200 °C, while Fig. 2d–f shows the results in the [110] projection. The imaging results at other temperatures are given in Supplementary Fig. 4. In addition to observing the structural changes in these images (the characteristics we discussed in Fig. 1), the intensity profiles in Fig. 2g, h, which are extracted from the red and blue arrows in Fig. 2a–f, also help to quantitatively describe these changes. In Fig. 2g and Supplementary Fig. 5, we defined two Cs-I peak distances as $L_1$ and $L_2$, the ratio of which indicates the offset of ion columns as discussed in Fig. 1e, h. In Fig. 2h, $L_3$ and $L_4$ were also defined for the same purpose.

As we have mentioned above, the phase transition in $CsPbI_3$ is related to the rotation of the $PbI_6$ octahedron. Based on the schematics and models of the phase transition in Fig. 2i, it is easy to infer that in the α phase, $L_1 = L_2$, $L_3 = L_4$, and $\theta_1 = \theta_2 = 0$. From the α to β phase, a rotation in the (001) plane occurs, and the $L_1/L_2$ ratio and $\theta_1$ begin to change. From the β to γ phase, another rotation out of the (001) plane occurs, and the $L_3/L_4$ ratio and $\theta_2$ begin to change. Therefore, in

Fig. 2j, k, we use these four parameters (the statistical results of 50 ratios and angles for each point) to quantitatively describe the change in phase in $CsPbI_3$. As the temperature rises, both the $L_1/L_2$ and $L_3/L_4$ ratios approach 1, and both $\theta_1$ and $\theta_2$ approach 0. More specifically, based on the data in Fig. 2j, k, $CsPbI_3$ is γ, β, and α phases in the temperature ranges of R.T to 150 °C, 150 to 250 °C and over 250 °C, respectively. The real-space imaging not only provides the structures before and after the phase transition but also shows the "intermediate states" during the phase transition. For example, the measured $\theta_1$ at 200 °C and $\theta_2$ at 100 °C are obviously the angles in the middle of those of complete phases. It is difficult to directly observe the change in rotation angle from the diffraction results, not to mention the quantitative study to obtain these angle values. On the one hand, these intermediate rotation angles indicate that the phase transition does not occur abruptly, and such structures between three phase models have not been fully considered in the performance studies and simulations previously. On the other hand, the error bars (standard deviations) in Fig. 2j, k demonstrate that a nonuniform spatial distribution of the rotation angle (that is, the spatial distribution of the phase) may also occur in each nanocrystal (see the histograms in Supplementary Fig. 6). These unexpected results about the phase transition and distribution in MHPs will bring in-depth thinking on the structural evolution between different phases of MHPs.

## Perovskite phase transition dynamics in single particle

It is worth noting that the images in Fig. 2 were captured after the sample was stable for over 1 h at each temperature, so that they only recorded the structures at the end of the phase transition at each temperature. To study the evolution process of phase structures (that is, the phase transition dynamics), we continuously captured three images at each temperature (100 and 200 °C) with a time interval of only one and a half minutes in a single nanocrystal. The raw images are given and numbered in Supplementary Fig. 7. Among them, figures i–iii are obtained at 100 °C, and figures iv–vi are obtained at 200 °C. Due to the short time interval, the phase transition recorded in these images was not completed, which leads to an obvious phase distribution (expressed as $\theta_1$ in the [001] projection) in this nanocrystal. For example, Fig. 3a is the iDPC-STEM image (figure vi in Supplementary Fig. 7) of this nanocrystal at 200 °C viewed from the [001] direction, and the magnified image in Fig. 3b is extracted from the black ribbon box in Fig. 3a showing the whole structure from the center to the surfaces of this nanocrystal. Comparing the atomic arrangement at the center (marked by the yellow box) and near the surfaces on both sides (marked by red and blue boxes) in Fig. 3c, it is obvious that the rotation angles of $PbI_6$ octahedrons at the center are larger than those near the surfaces, showing a continuous distribution of the MHP phase, probably from γ (center) to near α (surface).

Figure 3d shows the clear rotation of the same $PbI_6$ octahedron column in this series of images during heating. The left three panels indicate the change at 100 °C with time (from figures i–iii in Supplementary Fig. 7), and the right three panels follow at 200 °C (from figures iv–vi in Supplementary Fig. 7). The rotation angle $\theta_1$ reduced from 13.50 to 6.84° as marked by the purple boxes. The intensity profile analysis in Fig. 3e (extracted along the purple arrow) also shows the shift of the I peak with timeline to express such rotation (similar to the analysis in Figs. 1 and 2). Then, we use a three-dimensional plot in Fig. 3f to display the change in rotation angle with time and space at the same time. It provides a statistic on the thermal-induced rotation of $PbI_6$ octahedrons in each layer from the surface to the center. On the one hand, we can observe the decrease of rotation angle after heating, especially near the surface. The results of figures i–iii show the phase transition at 100 °C within 3 min, and more obvious changes at 200 °C are extracted from figures iv–vi. On the other hand, we can observe a clear phase distribution in this nanocrystal, which is expressed by the distance away from the surface, especially at higher temperatures. This

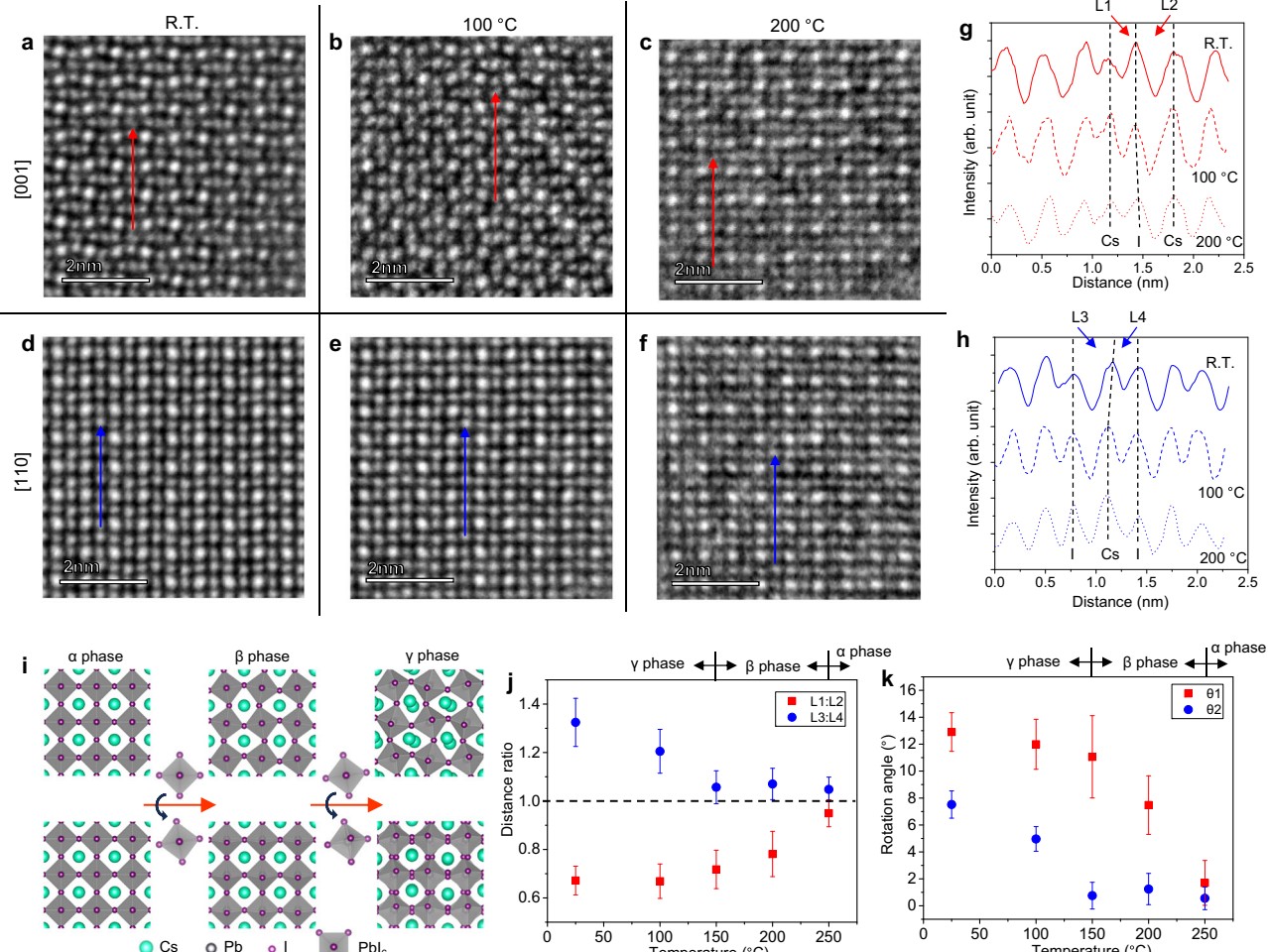

**Fig. 2 | Atomically resolving the changes in CsPbI₃ phase structures with temperature. a–f** IDPC-STEM images of CsPbI₃ nanocrystals at different temperatures (R.T., 100 and 200 °C) viewed from the [001] (**a–c**) and [110] (**d–f**) directions, respectively. **g, h** Intensity profiles extracted along the red arrows in (**a–c**) and the blue arrows in (**d–f**) to distinguish the phase structures at R.T. (solid lines), 100 (dash lines) and 200 °C (dot lines), respectively. The Cs-I peak distances, $L_1$, $L_2$, $L_3$, and $L_4$, are defined to quantitatively describe the positions of ion columns in different phases. **i** Schematics and models of the phase transition corresponding to the rotation of the PbI₆ octahedron. **j** Statistical results of the $L_1/L_2$ and $L_3/L_4$ ratios measured in the profiles at different temperatures. **k** Statistical results of $\theta_1$ and $\theta_2$ measured in the images at different temperatures. Source data are provided as a Source Data file.

indicates that the thermal effect gradually penetrated into a nanocrystal rather than occurring simultaneously in all parts of it. A penetration depth of only 5 nm was observed in Fig. 3f, while there was no phase transition or rotation in the central part within the time range we investigated.

Furthermore, we provided in situ heating XRD study in Supplementary Fig. 8. On the one hand, the small size of nanocrystals (10–30 nm) induces a much wider XRD peak of perovskite phase and adjacent characteristic peaks were fused into one wide peak. Only the slight shifts of these wide peaks proved that the phase transition from γ to α phase seems to have occurred. On the other hand, the non-perovskite phase (δ phase) was formed at over 120 °C, which makes it difficult to identify the shifts of perovskite peaks to study the phase transition at these temperatures by the XRD method. However, thanks to the localized nature of imaging methods, we can find the particles that are still perovskite phases at high temperatures to study the characteristics of phase distribution and obtain data on phase transition and its dynamics. This is why we believe that the real-space imaging method plays an irreplaceable role in studying this issue. Meanwhile, it was also reported that the phase transition temperature decreased with the reduction of nanocrystal size through the XRD in previous studies[47,48]. For the nanocrystals with 10–30 nm sizes in this work (as shown in Supplementary Fig. 9), the change in phase

transition temperature is not significant (within 20 degrees). It is reasonable to ignore the influence of size effect on phase transition temperature in our study.

Based on these results, we can obtain an in-depth understanding of the MHP phase transition dynamics. Such a phase transition in MHPs can occur continuously on a macro time scale (15 min in total). Figure 3g shows the in-situ changes in the rotation angles of three PbI₆ octahedron columns at different positions extracted from Supplementary Fig. 7. This result is consistent with the statistics in Fig. 3f, that is, the PbI₆ octahedrons near the surfaces are more affected by the heating process. According to the three sets of data at 100 °C and 200 °C in Fig. 3f, we can obtain the average speed of rotation of PbI₆ octahedrons at different distances away from the surface (Fig. 3h), which further provides a quantitative description of the phase transition dynamics in MHPs. The speed of rotation reaches the highest value of approximately 0.9°/min near the surface and decreases as the imaged position penetrates into the nanocrystal. The transition process is unexpectedly slow even at 200 °C, which can be exactly recorded using these iDPC-STEM images with a temporal resolution limited by the time interval between image acquisition. This inspired us to consider how these "intermediate state" structures and phase distribution (penetration depth) affect the intrinsic properties of both nanocrystals and films in theoretical simulations and real applications,

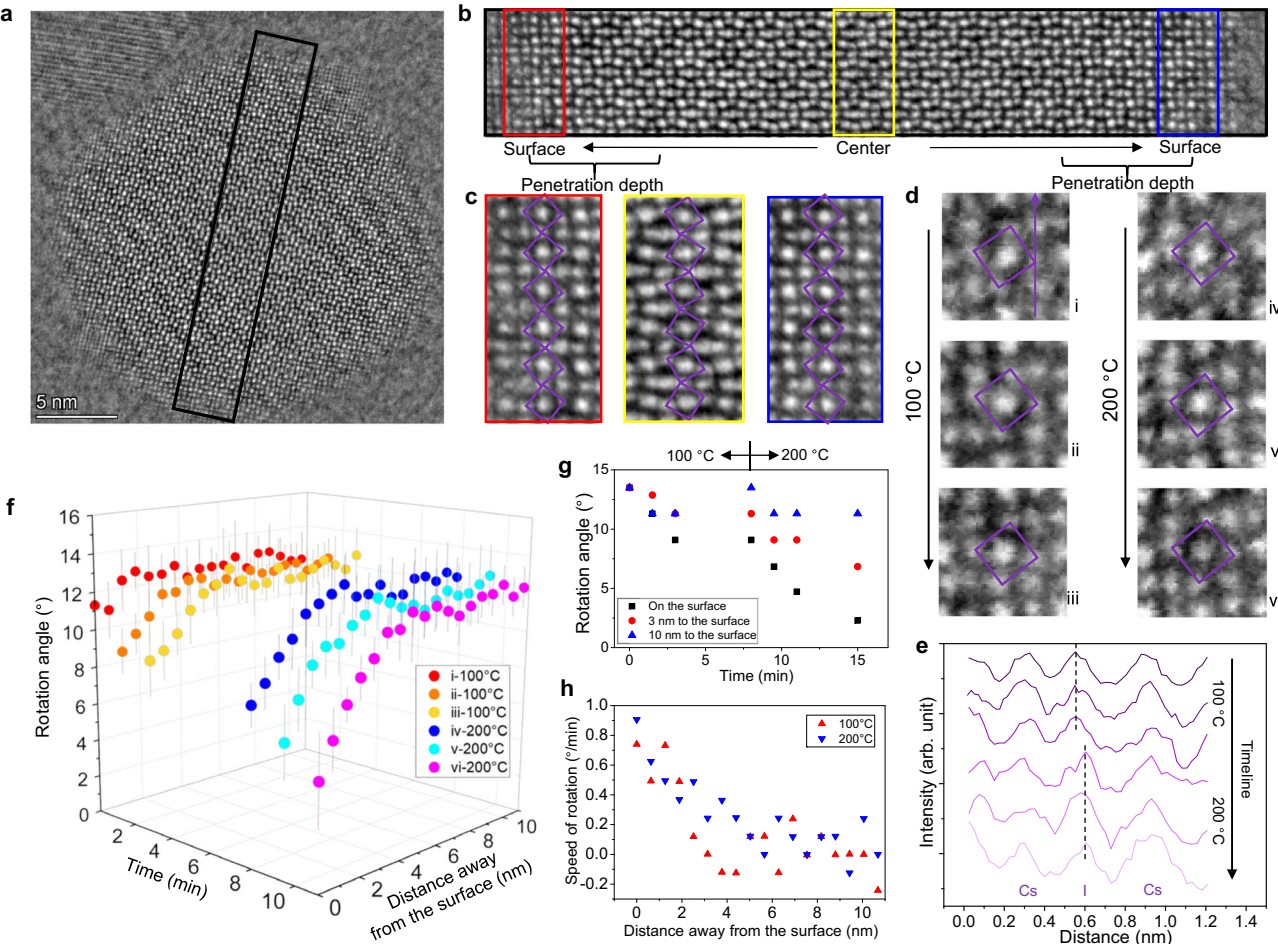

**Fig. 3 | In-situ imaging of the phase transition dynamics in a single CsPbI₃ nanocrystal. a** IDPC-STEM image of a CsPbI₃ nanocrystal viewed from the [001] direction (figure vi in Supplementary Fig. 7). **b** Magnified image extracted from the black box in (**a**) showing the whole structure from the center to the surface of this nanocrystal. **c** Three regions extracted from the colored boxes in (**b**) indicating the different rotation angles at the center and the surface. **d** Images extracted from the same area in different images (figures i-vi in Supplementary Fig. 7) showing the rotation of a PbI₆ octahedron column at 100 and 200 °C. **e** Intensity profiles extracted along the same direction marked by the purple arrow in **d** to distinguish

the phase structures in each image in (**d**). **f** Changes in the spatial distribution (expressed by the distance away from the surface) of rotation angle θ₁ within 3 min at 100 and 200 °C, respectively. Time stamps correspond to image acquisition interval in Supplementary Fig. 7. Figures i–iii were obtained continuously at 100 °C, while figures iv-vi show the results at 200 °C. **g** Rotation angles of PbI₆ at different positions extracted from Supplementary Fig. 7. **h** Speed of PbI₆ rotation at different positions calculated based on the data in (**f**). Source data are provided as a Source Data file.

since the time scale of a complete phase transition is comparable to that of our processing, characterizing and testing materials.

Then, we used the density functional theory (DFT) calculations to show how these observed phases affect the properties and stabilities of perovskite nanocrystals, as shown in Supplementary Fig. 10. The densities of states (DOS) of different phases of CsPbI₃ indicate a reduced band gap from the γ to α phase. The lowest potential energy of the γ phase of CsPbI₃ explains why it exists dominantly at room temperature. The transition structure we discussed in Fig. 3 was also optimized. Based on the band gap calculations, it is worth noting that series-connected type II heterojunctions are formed between the transition region of α to γ phase, which may effectively reduce the electron-hole pair recombination and improve the photoelectric conversion efficiency. In further study, it will be very interesting to discover corresponding new properties in such a transition phase.

## Degradation and conversion of CsPbI₃ particles

The above discussions atomically unraveled the phase transition process of CsPbI₃ itself under the condition of in-situ heating. However, we should also pay attention to the material changes at the particle level, which is also important for material processing, device preparation,

and stability. As shown in Fig. 4a, accompanied by the phase transition of CsPbI₃, the cubic CsPbI₃ nanocrystals began to lose their regular shape and gradually fused together at temperatures over 200 °C. Then, at temperatures over 300 °C, the change in particle morphology is more dramatic, and eventually, CsPbI₃ decomposes and is obviously converted into other substances. More images describing this evolution process of particles are shown in Supplementary Fig. 11. At 200 °C, we observed that the flat (110) surfaces in some cubic particles gradually disappeared, and more (010) surfaces were exposed (Supplementary Fig. 12). Figures 4b, c show the images of the same particle captured with a time interval of 1 min at 200 °C. The red and blue lines outlined the (010) surfaces before and after. The region between the two lines clearly indicates that one or two layers of PbI₆ octahedrons were released on the (010) surface. After that, at 200 to 300 °C, they gradually fused into large but unregular-shaped particles, which should be non-perovskite CsPbI₃ phase (δ phase) according to the in situ XRD results in Supplementary Fig. 8, although they cannot be atomically imaged due to their unregular shapes.

From 300 to 400 °C, the sample underwent a melting-like process, where the cubic CsPbI₃ particles quickly disappeared, and some larger elliptical particles were generated. Based on the elemental

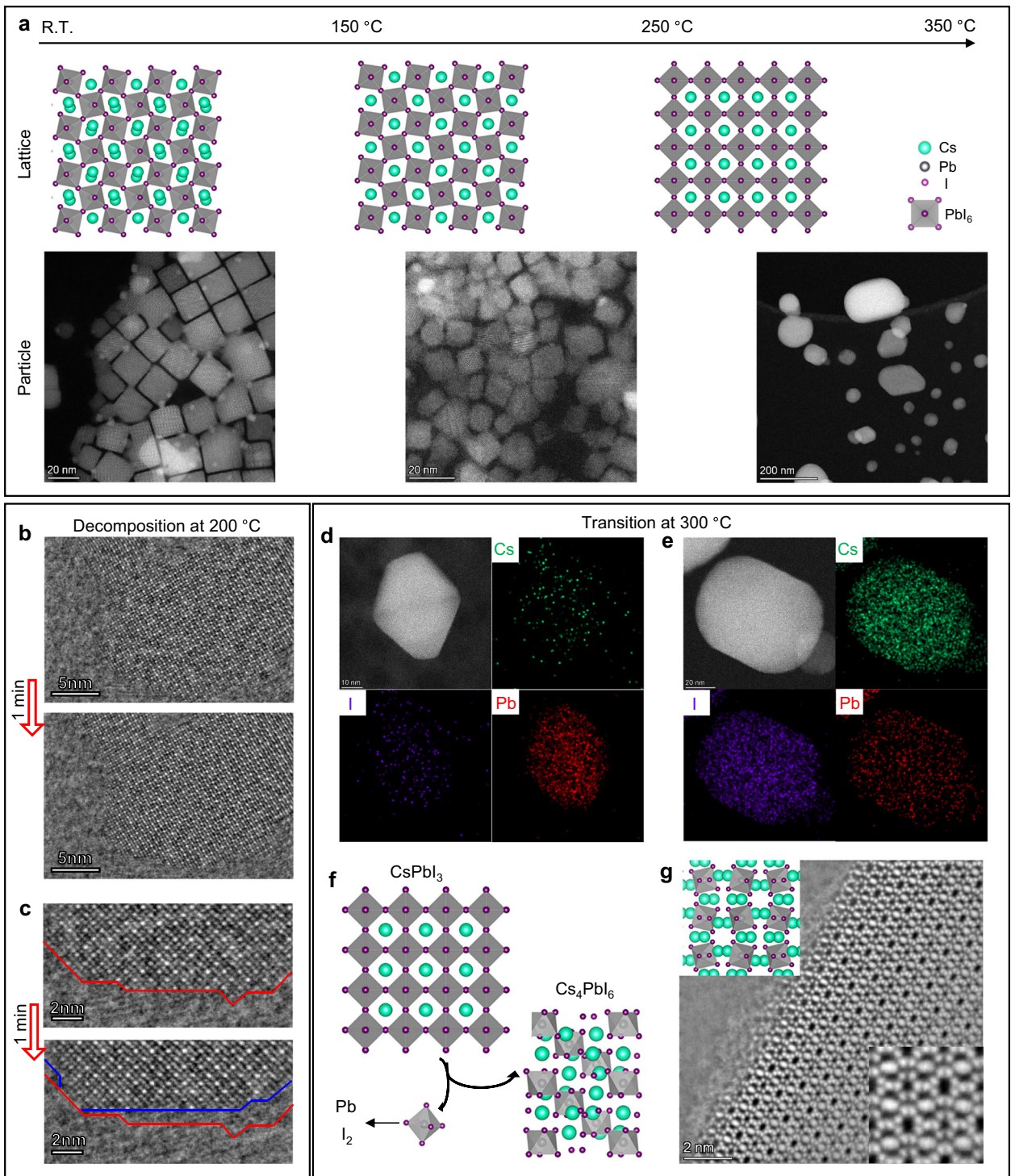

**Fig. 4 | Imaging the degradation and conversion of CsPbI₃ nanocrystals at the particle level. a** Schematics of the thermal-induced change in the lattice and particles of CsPbI₃ samples. The HAADF-STEM images show the changes in the sizes and shapes of CsPbI₃ nanocrystals with temperature. **b**, **c** IDPC-STEM images showing the surface degradation of a CsPbI₃ nanocrystal within 1 min at 200 °C. **d**, **e** EDS elemental mapping results of two particles generated at 300 °C indicating Pb and Cs₄PbI₆ particles (according to the relative proportions of elements), respectively. **f** Schematics of the conversion from CsPbI₃ to Pb and Cs₄PbI₆ at high temperature. **g** IDPC-STEM image of Cs₄PbI₆ particles viewed from the [211] direction to further confirm the generation of Cs₄PbI₆.

mapping by energy dispersive spectroscopy (EDS), it can be inferred that the products of material conversion at 300 °C should be pure Pb particles (Fig. 4d) and Cs₄PbI₆ particles (Fig. 4e), while only Pb can be found at 400 °C as shown by the EDS mapping in Supplementary Fig. 13. Thus, we can give a schematic structural evolution from CsPbI₃ to Pb and Cs₄PbI₆ in Fig. 4f. Moreover, we also used the iDPC-STEM to finally confirm the atomic structures of Pb (Supplementary Fig. 14) and

Cs₄PbI₆ (Fig. 4g and more images in Supplementary Fig. 15). The critical temperature (300–400 °C) is much lower than that in the in situ XRD results in Supplementary Fig. 8. Such degradation may be accelerated by the reduction of Pb²⁺ under electron beam irradiation and the volatilization of I₂ in vacuum. Although the electron dose for the iDPC-STEM was low enough not to damage the CsPbI₃ sample (as shown in Fig. 3), the influence of the electron beam in other imaging

modes, such as the HAADF-STEM images in Fig. 4a, cannot be completely ignored. Anyway, these results are important supplements to the whole structural evolution in MHPs at the particle level and, together with the observations of phase transition, provide comprehensive insight into the influence of temperature on MHPs.

In summary, with the progress of low-dose electron microscopy, the in situ structural changes in MHPs, including the phase transition and conversion, can be directly observed in real space. The iDPC-STEM can provide atomic insights into the temperature-dependent structural evolution. The transition from the γ to α phase of MHPs was quantitatively analyzed using the rotation of the $PbI_6$ octahedron in multiple dimensions, including the measurement of the rotation angle and mapping of its spatial distribution. Such in situ phase transition in a single nanocrystal can be studied by continuously imaging the rotation of the same $PbI_6$ octahedron, which makes it possible to unravel the phase transition dynamics with a proper spatial and temporal resolution. Our findings in this work, including the low rotation speed and obvious distribution during phase transition, have not been investigated and considered in previous studies. Therefore, directly "seeing" phase transition dynamics by electron microscopy is of great significance for explaining the phase-property relation in materials and improving the lifetime and stability of devices. Meanwhile, although the spatial resolution of the iDPC-STEM imaging has reached the atomic level, the temporal resolution still needs to be improved by using faster electron detectors to get more details of the phase transition. After further improving the imaging stability, it is also worth anticipating that the effects of other working environments (combining gas atmosphere and external field) on MHPs can be realized by more in situ imaging techniques. This work not only reveals the long-term unclear dynamics of the thermal-induced phase transition in MHPs but also provides more hope and confidence for exploring more physical and chemical phenomena in MHPs at the atomic scale.

## Methods

### Synthesis of CsPbI$_3$ nanocrystals
One gram of $Cs_2CO_3$, 4 mL oleic acid (OA) and 50 mL 1-octadecene (ODE) were added into a 250 mL three-neck flask and degassed under vacuum at 90 °C for 60 min. After that, the flask was filled with $N_2$ and heated to 120 °C until the $Cs_2CO_3$ and OA completely reacted to form a transparent Cs-oleate (CsOA) solution, and then stored in an $N_2$ glove box at 80 °C. Meanwhile, 1 g $PbI_2$ and 50 mL ODE were added into another 250 mL three-neck flask under vacuum at 90 °C for 60 min. Then, 5 mL OA and 5 mL oleylamine (OAm) were injected while purging the flask with stable $N_2$ flow. Under the protection of $N_2$, the mixture solution was heated to 160 °C until the $PbI_2$ was completely dissolved. Afterward, 4 mL of preheated CsOA was rapidly injected into the flask, and the reaction was quenched by an ice bath after approximately 5 s.

The crude solution was loaded equally into six centrifuge tubes. Subsequently, each CsPbI$_3$ nanocrystals solution was added to 32 mL anhydrous methyl acetate (MeOAc) (nanocrystal solution: MeOAc is 1:3) and then centrifuged at 7104 g for 5 min. The precipitate in each centrifuge tube was redispersed in 3 mL hexane, with 3 mL MeOAc (nanocrystal solution: MeOAc is 1:1) added, and centrifuged at 7104 g for 3 min. Then, the obtained nanocrystal precipitate was redispersed in 20 mL hexane and centrifuged at 1776 g for 5 min to remove excess $PbI_2$ and CsOA. Finally, the CsPbI$_3$ nanocrystal solution was stored at 4 °C before the iDPC-STEM imaging.

### Atomic imaging by electron microscopy
A Cs-corrected scanning transmission electron microscope (FEI Titan Cubed Themis G2 300) equipped with a DCOR+ spherical aberration corrector for the electron probe was operated at 300 kV and $10^{-5}$ Pa to obtain all the iDPC-STEM and HAADF-STEM images. During the iDPC-STEM imaging, the convergence semi-angle is 15 mrad. The collection angle is 7–36 mrad. The beam current is approximately 0.1 pA (measured by a pixel array detector). The dwell time for each pixel is 32 μs. The pixel size is 0.1257 Å. Therefore, the electron dose for iDPC-STEM is approximately 1266 e$^-$/Å$^2$. During the imaging process, the concentrated perovskite solution was dropped onto ultra-thin carbon film. The CsPbI$_3$ nanocrystals were not artificially oriented, but they are all cubic or cuboid, so that one of three orthogonal directions of a cubic nanocrystal may align perfectly with the electron beam and the atomic arrangement in this projection can be clearly imaged.

The image simulations for the iDPC-STEM were conducted based on the multi-slice method[49–51]. The parameters for the image simulations are the same as those in the experiments. The heating experiment was conducted using an in situ holder (Protochips, Fusion 350), which was heated to different temperatures with an ultrahigh heating rate of 1000 °C/ms (that is, immediately). All the intensity profiles were extracted with only one-pixel width, and the intensity of each point just represents the intensity of this pixel. Therefore, there is no averaging involved perpendicular to profile direction.

### In situ XRD study
The in situ XRD samples are prepared by dropping the concentrated perovskite solution onto the center of a Si substrate, and then the diffraction patterns were obtained with Bruker D8 Discover (Cu Kα radiation, λ = 1.5418 Å). In detail, the in situ heating XRD analyses were performed with a heating speed of 10 °C/min and a duration time of 600 s from 25 °C to 600 °C under a flowing nitrogen atmosphere.

### Computational methods and models
All calculations were carried out using VASP (Vienna Ab-initial Simulation Package, VASP) software 5.4.4[52–54], the exchange function was described by PBE (Perdew, Burke, and Ernzerhof) functional parameterization of GGA (Generalized Gradient Approximation, GGA) of DFT (Density of Functional Theory)[55]. These calculations took DFT-D3 correction for London disperse with Becke-johnson damping into consideration[56]. The PAW (Projector Augmented Wave) method was used to account for core-valence interactions[57,58]. The kinetic energy cutoff for plane wave expansions was set 400 eV, and reciprocal space was sampled by Γ-centered Monkhorst–Pack scheme with a grid of 8 × 8 × 8 for cell optimization, 3 × 3 × 1 self-consistent computation and K path for energy calculations set by VASPKIT[59]. The convergence criteria are $1 \times 10^{-5}$ eV energy difference for solving the electronic wave function. All atomic coordinates were converged within $1 \times 10^{-2}$ eV Å$^{-1}$ for maximal components of force. Next, we built models and started DFT calculations over α, β, and γ phase CsPbI$_3$ for studying electronic property and stability, and the transition structures built by packing CsPbI$_3$ (001) of α phase and CsPbI$_3$ (110) of γ phase along the Z-axis, the former structure was modeled by 2 × 2 × 2 unit cell, the later one was modeled by 2 × 2 × 3 unit cell without vacuum, the extra I atoms in the interface were removed and translated the position of α phase CsPbI$_3$ by 0.9 Å alone y-axis conformed to results of TEM (Transmission Electron Microscope). On the structural optimization, the α phase in the interface structure has been fixed and half of the γ phase in the left area has been also fixed to keep the Pb-I octahedron unchanged.

## Data availability
The authors declare that all relevant data supporting the findings of this study are available in the paper and its Supplementary Information files or from the corresponding authors upon request. Source Data file has also been deposited in Figshare under the accession link https://doi.org/10.6084/m9.figshare.24441322[60]. Source data are provided with this paper.

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

## Acknowledgements

This work was supported by the National Natural Science Foundation of China (T2322019 (B.S.), 22275133 (B.S.), 52261145696 (J.Y.)), Suzhou Science and Technology Development Plan (ZXL2023179 (B,S.)), National Key Research and Development Program of China (2019YFE0108600 (J.Y.)), Science Foundation of Jiangsu Province (BK20220484 (B.S.), BK20211598(J.Y.)), Suzhou Key Laboratory of Functional Nano & Soft Materials, Collaborative Innovation Center of Suzhou Nano Science & Technology, the 111 Project, Joint International Research Laboratory of Carbon-Based Functional Materials and Devices.

## Author contributions

B.S. conceived this project and designed the studies; M.M. wrote and revised the manuscript; B.S., F.W., and X.C. performed the imaging experiments; B.S. and M.M. analyzed the imaging data and wrote the manuscript; X.Z. and J.Y provided the MHP samples; H.X. performed the image simulation; L.X. and T.C. carried out the DFT energy calculations; All authors are involved in the data analysis.

## Competing interests

The authors declare no competing interests.
