## [Peer Review File · Nature Communications]

REVIEWER COMMENTS

Reviewer #1 (Remarks to the Author):

The authors explore the phase transition phenomenon of metal halide perovskites through the use of in-situ iDPC-STEM. Atomic resolution images of CsPbI₃ across various phases and temperatures illustrate the phase transition as temperature changes. The phase distribution within a single particle was also assessed. However, I question the use of the term "Dynamics" in the title, as there doesn't seem to be substantial high-temporal-resolution atomic dynamical information provided during the phase transition, aside from some low-temporal-resolution data in Extended Data Fig.4. Furthermore, for this manuscript to meet the standards for publication in Nat. Commun., several other concerns need to be addressed.

1. There are numerous pre-existing (S)TEM studies on CsPbI₃ nanocrystals. Atomic-resolution imaging of this specific material isn't particularly challenging. The authors claim that iDPC-STEM is a low-dose technique suitable for in-situ imaging of this material. However, the main text doesn't discuss the exact electron doses utilized. It remains unclear why iDPC-STEM would be more advantageous than other (S)TEM techniques in this instance. For instance, HAADF-STEM can provide very high-quality images for CsPbI₃ nanocrystals. The authors should compare HAADF-STEM and iDPC-STEM in terms of dose requirement, image contrast, and information obtained, to illustrate the necessity and advantages of using iDPC-STEM.
2. Despite the extensive study of temperature-dependent phase transition of CsPbI₃, it remains essential to conduct bulk characterization, such as in-situ heating PXRD, for the sample used in this study. This will help reinforce and validate the conclusions drawn from the imaging. After all, atomic-resolution imaging, due to its highly localized nature, may not be fully representative of the entire sample.
3. The electron beam effect should be carefully discussed. Experiments should be conducted to demonstrate whether or not long exposure to the electron beam can induce phase transition without heating?
4. The authors should discuss the reversibility of the phase transition process. For instance, if the temperature decreases from 200 °C to room temperature, will the β phase revert back to the γ phase?
5. The authors have exaggerated or stated inaccuracies in many places. For example, in "Introduction", they stated "However, these previous studies can only provide the averaged structural information of macro samples without any temporal and spatial resolution..." However, they said later that "These findings are consistent with our understanding of the phase transition of MHPs based on in-situ X-ray diffraction¹⁹." This is a clear contradiction! The in-situ XRD method can certainly provide temporal resolution.
6. Another example, the authors claimed that "we developed the integrated differential phase contrast (iDPC)..." iDPC-STEM was NOT developed by the authors.
7. Besides the elemental mapping, provide the counts (spectrum of the elements) of the EDS results.
8. Scale bars in Fig. 2a-f, Fig.4b, c, Fig. S2 are missing.
9. Error bars in Fig. 3g, h, Fig. S3 should be provided.
10. The level of English in this manuscript is unsatisfactory. Some phrases appear awkward. For instance, the repetitive use of "unravel" in the sentence "... are unraveled by continuously imaging a single quantum dot to unravel the dynamics of the..." should be revised. I recommend the authors utilize tools like ChatGPT to help improve the language of the manuscript.

Reviewer #2 (Remarks to the Author):

The author reports the phase transition process of metal Halide perovskites by means of low-dose imaging technique, IDCP. Especially, the rotation angles of PbI₆ octahedrons can be measured in these images to quantitatively describe the thermally induced phase distribution and phase transition. It provides new atomic insights into the transition dynamics of perovskite phases, indicating a long-time transition process with obvious intermediate states and spatial distribution. The results are interesting and are important for the structure-property relation. But it cannot be

accepted before some revision. Some comments and questions are as follows.

Major revision:

- 1 Only structural variation is focused, and more discussions need to be added to enrich the manuscript, such as structure-property relation, or from the point view of energy
- 2

Minor revision

- 1 Structural models in Fig. 1 should be introduced in detail to make it clear, for example which balls represent Cs and P and I atoms.
- 2 It is not sure that the orthogonal surfaces correspond to the [001] and [110] directions as shown in Fig. 1b, for there many directions perpendicular to the [001], such as [100] and [010].
- 3 What happen for the figs. 2 c and f, in which there are more noise than other ones
- 4 I am confused whether the sample will transform to other phase after heating to 250°C. If so, and all these structure will be compared as shown in Figs. j and k, the data on 250°C are suggested to be shown in the manuscript other than in the extended data.

Reviewer #3 (Remarks to the Author):

The manuscript from Shen et al. provides an interesting insight into all-inorganic lead halide perovskites phase transitions, focussing in particular on CsPbI₃ (which is indeed the less stable member of the family, both chemically and in terms of 3D corner-sharing perovskite phase). The idea behind the article is in principle of high relevance for the nanoscience and material scientist community, but some fundamental/methodological points need to be clarified before considering the article for publication:

1) One of the main findings of the work is the possibility of mapping the octahedra tilting angles (Θ_1 and Θ_2) in CsPbI₃, upon T- variation. First, I would suggest removing from the article the claim that is not feasible to distinguish changes in these angles from X-ray diffraction (any structural paper on Lead halide perovskites from single crystal to powder diffraction, report the two distinct angles for the orthorhombic gamma phase). Second, it is not clear to me how the authors have properly oriented the nanocrystals under the electron beam, to be able to "label" the crystallographic directions that they claim to map ([001] and [110]): in the articles, I cannot find any detail about the strategy adopted by the authors in this sense; how are they sure that they are looking at that specific crystallographic directions? The two/three crystallographic directions in lead halide perovskites are very similar in terms of lattice periodicity, so it's very difficult to distinguish between them.

Moreover, it's not clear to me how the "penetration" studies of the electron beam have been performed. How is it possible to separate the influence from the surface/core in the nanocrystals analyzed? I suppose that the electron beam has a certain penetration depth and what is mapped is an average structure within this portion of nanocrystals analyzed.

2) 20 nm nanocrystals cannot be considered Quantum dots (as it is stated in the article). With this nanocrystals size, we are far away from quantum-confinement (being the Bohr radius for these materials something close to 4-5nm), and (on the structural and thus on the point of view of the electronic properties) very close to the bulk material; therefor the classical phase transition scheme from the gamma to the alfa phase reported in this work is not a huge novelty, being already reported for lead halide perovskites bulk materials. The novelty can be the methodology used, but it must be properly described in the manuscript.

Also, the shape of the nanocrystals analyzed is not clearly described (if isotropic makes it really difficult, I would say impossible, to properly orient the different crystallographic directions with respect to the electron beam).

3) CsPbI₃ bulk is not stable at ambient conditions in the perovskite gamma phase but it rapidly transforms to another orthorhombic phase, which is a not-perovskitic one (the so-called "yellow" phase). Have the authors considered the possibility of having this phase at room temperature? What would their simulation/intensity profiles look like in that case?

4) The definition of L1-L4 is not very clear: from what I've understood from Figure 2 the authors have considered a single Cs-I peak separation to compute these lengths. Or several values have been averaged within a single (or even multiple) nanocrystals?

5) How many nanocrystals have been analyzed? Do they show the same results in terms of phase transition and octahedra rotations?

6) The use of crystallographic directions throughout the article is done without clearly stating which is the reference system.

Reviewer #4 (Remarks to the Author):

Shen et al. report on spatially and temporally resolved phase transitions in single CsPbI₃ perovskite quantum dots (QDs). A major novelty of their work is the ability to resolve phase transitions in real space, with atomistic resolution, and to study such processes in single QDs, as opposed to more wide-spread scattering-based techniques which are thus far only ensemble-based. The key enabling element for this study was the combination of low-dose electron-microscopy with a novel imaging technique (iDPC-STEM, i.e. integrated differential phase contrast scanning transmission electron microscopy) recently developed by the authors. Application of this new technique also to perovskite QDs is a major step forward in understanding and controlling phase transitions and stability for this important class of materials which has already indicated promise in a wide range of applications, from solar cells to quantum-light sources. Overall, I suggest that this work is of significance for the broad readership of Nature Communications, after several revisions suggested here below.

The description of the new experimental technique has already been described in previous recent works of the authors. Hence, a main merit of the current manuscript is to apply this technique to a specific new material of interest, i.e. CsPbI₃ QDs. However, here the manuscript still shows some weakness as it sometimes fails to sufficiently compare the new findings with the current state-of-the-art knowledge in the perovskite QD community. For example, there is insufficient discussion of (1) how the here found phase transition temperatures compare to previously published values (2) how the phase transition depends on NC size; several previous studies have discussed the large reduction of phase transition temperatures with decreasing NC size, summarized e.g. in Mater. Adv., 2021, 2, 47. The authors may provide a QD size histogram in the SI and discuss ideally briefly in the main text) how their sizes and findings relate to size-dependent phase transitions in the literature.

(3) how the environment affects the phase transition; there is a large body of literature highlighting the importance of humidity, pressure, or strain due to different thermal expansion of material and substrate; the current work deals with a very specific and rather uncommon case, i.e. single QDs in vacuum, on a TEM substrate; a comment (best via referring to suitable previous literature) should be made how the reader can connect the found phase-transition here with "real-world" applications at ambient conditions in the applications mentioned by the authors in their intro.

(4) how to reconcile the slow phase transition rate found here (proceeding over several minutes) to the sub-nanosecond photoinduced phase transitions reported previously (see for the very related CsPbBr₃ QDs e.g. Nature Communications 2019, 10, 504, or for MAPbI₃ bulk e.g. Matter 2023, 6, 2, 460).

The discussed directions/projections seems to correspond to the pseudocubic system. This is important and, hence, should be mentioned and briefly discussed to

- clarify that those do not refer to the respective tetragonal and orthorhombic systems (which are different!)
- discuss the peculiar situation that the shown NCs seem to exhibit a surface termination along the pseudocubic axes, irrespective of the fact that they are orthorhombic. Is this representative for the studied NC batch? And how does this relate to other previous studies of CsPbI₃ NCs?

The experimental methods section in the SI is rather short. However, some more info would be useful for the reader, e.g. the precise vacuum and substrate employed.

The labelling of figures and captions is not always complete and sometimes even inconsistent and confusing.

- Fig. 1, 2, and 3: More details should be given on how exactly the intensity profiles were

obtained. Is some averaging perpendicular to the profile direction involved? This info may be given in the SI section, while the main text/caption can then refer to this section.

- Fig. 1c, d, f, g: the precise region from which (d) was extracted should be indicated in (c) with a box; same for (f) and (g). This is important as the surface-near QD regions behave differently than the QD interior, as shown later in the manuscript.

- Regarding inconsistent (and therefore confusing) color choice in Fig. 1:

- Fig. 1a: red and blue distinguish angles θ_1 and θ_2 , respectively

- Fig. 1d and 1g: red and blue indicate some line profile along different directions than those involved in defining the angles θ

- Fig. 1e and 1h: red and blue distinguish γ and β phase, respectively

These inconsistencies should be avoided and the color choice improved, to better guide the reader.

- Fig. 3: the labelling of subfigures and captions for subfigures (d) and (f) is confusing and could be improved. Furthermore, essential info (e.g. on timing within the temperature ramp) is omitted.

Few suggestions:

- do not use Fig. no. 1, 2, 3 (for the sub-sub-figures), as this is easily confused with Fig. 1, 2, 3 (reserved for the main figures). Use instead for example labels (i), (ii), and (iii). This means each panel can be unambiguously identified, as Fig. 3d(i), etc.

- The new labelling should be applied to both images in (d) and the respective axis in (f)

- Better still would be if the consecutive "index" labelling 1-6 would be replaced with actual time stamps (I presumed it should be 1.5 min, 3 min, 8 min, 9.5 min, 11 min, 15 min?) along with info on when (at 8 min?) the temperature was changed from 100 to 200 deg C. This would mean adding labels to the images in (d) and changing the labelling of the "time" axis in (f). All info in (d) and (f) should allow the reader to unambiguously connect to the data points shown in (g).

- Fig. 3 g,h: it should be specified which rotation angles is given, i.e. θ_1 or θ_2 .

RESPONSE TO REVIEWERS' COMMENTS

Reviewer #1:

Thank you for your time on our manuscript. We are glad to see your recommendation for our work. In your comments, you have raised some professional suggestions that can truly help us to make necessary supplement and improvement to this manuscript. We have made a careful revision based on the following replies to all your comments and suggestions. Meanwhile, we added in-situ heating PXRD for the sample used in this study and combined them with atomic-scale imaging results to better understand the phase transitions of CsPbI₃ perovskites. We sincerely ask you to consider our revised manuscript for further publication.

(1) The authors explore the phase transition phenomenon of metal halide perovskites through the use of in-situ iDPC-STEM. Atomic resolution images of CsPbI₃ across various phases and temperatures illustrate the phase transition as temperature changes. The phase distribution within a single particle was also assessed. However, I question the use of the term "Dynamics" in the title, as there doesn't seem to be substantial high-temporal-resolution atomic dynamical information provided during the phase transition, aside from some low-temporal-resolution data in Extended Data Fig.4.

Reply: Thank you for your comments. The concept of dynamics itself doesn't include limitations on temporal scale and resolution. The temporal resolution of the selected method for detecting a dynamic process should match its temporal scale. For example, as we found in perovskite system, the phase transition of CsPbI₃ is a slow dynamic process lasting at least 15 minutes from the γ phase to the α phase. If we use ultra-fast detection methods, we cannot observe any changes in structural information within an extremely small temporal scale. During our in-situ imaging process, the time interval of continuous acquisition of iDPC-STEM is about 1.5 minutes, which can just help us observe the appropriate degree of structural change. In other words, such temporal resolution is sufficient for us to study the slow dynamics of phase transition process of

perovskites, and higher temporal resolution is unnecessary. Meanwhile, in real-space imaging, temporal and spatial resolution are often contradictory to each other, that is, higher temporal resolution will sacrifice the spatial resolution, making it impossible to atomically identify such phase distribution and phase transition dynamics. This is also the reason why we chose this system for in-situ atomic imaging study: low temporal resolution is sufficient, while high spatial resolution can be guaranteed.

(2) There are numerous pre-existing (S)TEM studies on CsPbI₃ nanocrystals. Atomic-resolution imaging of this specific material isn't particularly challenging. The authors claim that iDPC-STEM is a low-dose technique suitable for in-situ imaging of this material. However, the main text doesn't discuss the exact electron doses utilized. It remains unclear why iDPC-STEM would be more advantageous than other (S)TEM techniques in this instance. For instance, HAADF-STEM can provide very high-quality images for CsPbI₃ nanocrystals. The authors should compare HAADF-STEM and iDPC-STEM in terms of dose requirement, image contrast, and information obtained, to illustrate the necessity and advantages of using iDPC-STEM.

Reply: Thank you for your comments. As you suggested, we provided the HAADF- and iDPC-STEM images of the same region in perovskite nanocrystal for comparison in Fig. R1. First, the electron dose for iDPC-STEM is about $1266 \text{ e}^-/\text{\AA}^2$, while that for HAADF-STEM is usually over $20000 \text{ e}^-/\text{\AA}^2$ (such high dose is required for enough signal-to-noise ratio). We cannot fully guarantee that the crystals were not damaged in high-dose HAADF-STEM images. For example, we can observe some bright spots in the HAADF-STEM image, which just represent the Pb particles precipitated from the crystals. Second, comparing HAADF- and iDPC-STEM images obtained at the same low dose, the contrasts between the light (I and Cs) and heavy (Pb) elements in the HAADF-STEM image are higher than those in the iDPC-STEM image, which makes that these light elements (I and Cs) cannot be clearly observed together with Pb. Especially, our study on the phase transition is mainly based on precise identification of the positions of Cs and I ion columns. Finally, the surface structure can be resolved

more clearly in the iDPC-STEM image due to obviously higher signal-to-noise ratio. Therefore, it should be more advantageous for us to study changes in CsPbI₃ structure and phase by using the iDPC-STEM. We have added these discussions and figures in the manuscript and supporting information as follows:

“When comparing the HAADF- and iDPC-STEM images collected simultaneously with the same low dose (about 1266 e⁻/Å²) in Supplementary Fig. 1, the light elements (I and Cs) and surface structures can be observed more clearly together with Pb in the iDPC-STEM image due to the imaging contrast linear to the atomic number during the integration process from four original images (Supplementary Fig. 2), which helps us to identify the ion position and other detailed structural information of each phase and local structure.”

Fig. R1 Comparison of HAADF- and iDPC-STEM images of the same region with the same low dose in CsPbI₃ nanocrystal.

(3) Despite the extensive study of temperature-dependent phase transition of CsPbI₃, it remains essential to conduct bulk characterization, such as in-situ heating PXRD, for the sample used in this study. This will help reinforce and validate the conclusions drawn from the imaging. After all, atomic-resolution imaging, due to its highly localized nature, may not be fully representative of the entire sample.

Reply: Thank you for your comments. As you suggested, we provided in-situ heating PXRD results in Fig. R2 (also Supplementary Fig. 8). In detail, the in-situ heating

XRD analyses were performed with a heating speed of 10°C/min and a duration time of 10 min from 25°C to 600°C under flowing nitrogen atmosphere.

In Fig. R2, on the one hand, the small size of nanocrystals (10-30 nm) induces a much wider XRD peak of perovskite (adjacent characteristic peaks were fused into one wide peak) compared with those in the simulated XRD results, so that it is difficult to get sufficient detailed diffraction information to clearly distinguish different perovskite phases. Only the slight shifts of these wide peaks (especially at 2theta of about 28.5°) can prove that the phase transition from γ to α phase seems to have occurred. On the other hand, the non-perovskite phase (δ phase) was formed at over 120 °C, and the peaks of perovskite phases were completely masked at over 220 °C. It is difficult to identify the peak shifts of perovskite phases to study the phase transition at these temperatures by the XRD method. However, thanks to the localized nature of imaging methods, we can find the particles that are still perovskite phases at high temperatures during electron beam scanning to study the characteristics of phase distribution and obtain data on phase transition and its dynamics, which cannot be separated from the averaged XRD data. This is why we believe that the real-space imaging method plays an irreplaceable role in studying this issue.

Meanwhile, we found that the degradation temperature (about 600 °C) is much higher than that observed by electron microscopy (about 400 °C). This phenomenon has been discussed in Fig. 4 in our manuscript. Such degradation may be accelerated by the reduction of Pb^{2+} under electron irradiation and the volatilization of I_2 in vacuum. Although the electron dose for the iDPC-STEM was low enough not to damage the CsPbI_3 sample, the influence of the electron beam in other imaging modes, such as the HAADF-STEM images in Fig. 4a, cannot be completely ignored. Then, the CsPbI_3 nanocrystals evolved into Cs_4PbI_6 structures at higher temperatures, as we observed in Fig. 4 by both EDS mapping and iDPC-STEM imaging.

Therefore, XRD method is an important supplement to the conclusions drawn from the imaging results. But it still has intrinsic limitations in characterizing the transition between perovskite phase, especially when the non-perovskite phase interferes with our judgment of perovskite peaks as an impurity. Combining two methods (diffraction

and electron microscopy) helps us better understand the whole structural evolution (including phase transition and decomposition) of materials in both the bulk and local structures. As you have suggested, we have added following discussions in the revised manuscript and in-situ XRD results in the supporting information.

“Furthermore, we provided in-situ heating X-ray diffraction (XRD) study in Supplementary Fig. 8. On the one hand, the small size of nanocrystals (10-30 nm) induces a much wider XRD peak of perovskite phase and adjacent characteristic peaks were fused into one wide peak. Only the slight shifts of these wide peaks proved that the phase transition from γ to α phase seems to have occurred. On the other hand, the non-perovskite phase (δ phase) was formed at over 120 °C, which makes it difficult to identify the shifts of perovskite peaks to study the phase transition at these temperatures by the XRD method. However, thanks to the localized nature of imaging methods, we can find the particles that are still perovskite phases at high temperatures to study the characteristics of phase distribution and obtain data on phase transition and its dynamics. This is why we believe that the real-space imaging method plays an irreplaceable role in studying this issue.”

Fig. R2 Analysis of the in-situ XRD results for CsPbI₃ nanocrystals. The data are collected at different elevated temperatures ranging from 25 °C to 600 °C and compared with the standard patterns of different phases.

(4) The electron beam effect should be carefully discussed. Experiments should be conducted to demonstrate whether or not long exposure to the electron beam can induce phase transition without heating?

Reply: Thank you for your comments. The damage to perovskites during imaging has been discussed in our previous published article (Adv. Mater. 2023, 202300653). In Fig. R3, we give four atomic-resolution iDPC-STEM images of CsPbI₃ nanocrystals taken continuously at room temperature with an interval of about 1-2 min between the two shoots, indicating that the bulk phase and surface structures of nanocrystals will not change under the electron irradiation in our imaging experiment. On this basis, we can ignore the effect of electron irradiation on the lattice structure of CsPbI₃ and focus on the thermal-induced phase transition we studied in this paper.

Fig. R3 IDPC-STEM images with atomic resolution of CsPbI₃ nanocrystals taken continuously at room temperature. The interval between the two shoots is about 1-2 minutes.

(5) The authors should discuss the reversibility of the phase transition process. For instance, if the temperature decreases from 200 °C to room temperature, will the β phase revert back to the γ phase?

Reply: Thank you for your comments. We cooled down the sample after heating to 200 °C to observe its structural change. It is confirmed that the CsPbI₃ nanocrystals reverted back to the γ phase as shown in Fig. R4, indicating reversible phase transition of CsPbI₃ nanocrystals. The decomposition and fusion between particles indicate that they have already gone through the heating process and then cooled down to the room temperature.

Fig. R4 Imaging the γ phase structure after cooling down to room temperature.

(6) The authors have exaggerated or stated inaccuracies in many places. For example, in “Introduction”, they stated “However, these previous studies can only provide the averaged structural information of macro samples without any temporal and spatial resolution...” However, they said later that “These findings are consistent with our understanding of the phase transition of MHPs based on in-situ X-ray diffraction¹⁹.” This is a clear contradiction! The in-situ XRD method can certainly provide temporal resolution.

Reply: Thank you for your comments. We have noticed that there may be an incorrect expression here. Our main purpose here is to emphasize the advantages of real space imaging methods in terms of spatial resolution. We have modified some of the statements in the Introduction as follows:

“However, these previous studies by the diffraction methods mainly provided the averaged structural information of macro samples without local structural information, limiting our understanding of atomic phase distribution and phase transition dynamics in MHPs, especially when the non-perovskite phase can act as an impurity to interfere with our positioning of diffraction peaks in perovskite phase.”

(7) Another example, the authors claimed that “we developed the integrated differential phase contrast (iDPC)...”. iDPC-STEM was NOT developed by the authors.

Reply: Thank you for your comments. As you suggested, we have changed the statement as follows in the revised manuscript:

“Recently, we used the integrated differential phase contrast (iDPC) scanning transmission electron microscopy (STEM) for beam-sensitive materials^{40,41}, and then, combined it with an in-situ imaging technique to realize the in-situ observation of the adsorption/desorption behaviors of small molecules⁴²⁻⁴⁵.”

(8) Besides the elemental mapping, provide the counts (spectrum of the elements) of the EDS results.

Reply: Thank you for your comments. The elemental mappings in Fig. 4 were used mainly to assist us in investigating the material structures. Since the materials studied in this work, especially the CsPbI₃ and Cs₄PbI₆ crystals, are very sensitive to the electron beam, long-time electron irradiation during EDS mapping can cause electron damage and sample drift. Therefore, the electronic dose used for counting in the EDS experiment can only be set to be lower, which does affect the accuracy of elemental quantification. Thus, the EDS results cannot be used as a quantitative description of the exact proportion of elements in these materials, but rather help us guess possible structural changes by a rough analysis on the relative contents of elements. However, this will not affect our structural confirmation, since it has already explained by direct atomic iDPC-STEM imaging as shown in Fig. 4g and Supplementary Fig. 14 and 15. That is also why we didn't include quantitative EDS data and counts in the figures in the main manuscript, which cannot add new understanding and necessary conclusions, especially when the size of Fig. 4 is already too large in the manuscript. We have also provided a more conservative explanation and discussion on the role of EDS mapping in the revised manuscript and hope you can understand our explanation here.

(9) Scale bars in Fig. 2a-f, Fig.4b, c, Fig. S2 are missing.

Reply: Thank you for your comments. We have added the missing scale bars in these figures.

(10) Error bars in Fig. 3g, h, Fig. S3 should be provided.

Reply: Thank you for your comments. Fig. 3g described the rotation of single PbI_6 octahedral column with time. Three sets of data points (black, blue and red) indicate the rotation angle of three PbI_6 octahedral columns in different regions (on surface, 3 nm to surface and 10 nm to surface) extracted from Supplementary Fig. 7. Therefore, each data point represents one measured rotation angle of this PbI_6 octahedral column instead of the statistical results of multiple measurements, and there shouldn't be error bars here. Meanwhile, the rotation speed at each position in Fig. 3h is also calculated from the value of one PbI_6 octahedral column instead of the statistical results.

As for the histograms in Supplementary Fig. 6, they provide the counts of different rotation angles, showing the distributions of θ_1 and θ_2 at different temperatures. The counts mean the total quantity of the whole, and there seems no concept of error for the counts. The error bars in Fig. 2k are just derived from these counts.

Based on the above reasons, we think that there should be no error bars or there is no need to add error bars in these three figures.

(11) The level of English in this manuscript is unsatisfactory. Some phrases appear awkward. For instance, the repetitive use of "unravel" in the sentence "... are unraveled by continuously imaging a single quantum dot to unravel the dynamics of the..." should be revised. I recommend the authors utilize tools like ChatGPT to help improve the language of the manuscript.

Reply: Thank you for your suggestion. We have used language polishing tools to check and correct grammatical and stylistic errors in the text.

Reviewer #2:

Thank you for your time on our manuscript. We are glad to see your recommendation for our work. In your comments, you have raised some professional suggestions that can truly help us to make necessary supplement and enrich our manuscript. We have made a careful revision based on the following replies to all your comments and suggestions. Meanwhile, we added the density functional theory calculations to better understand the relationship between structures and the properties of perovskites. We sincerely ask you to consider our revised manuscript for further publication.

(1) Only structural variation is focused, and more discussions need to be added to enrich the manuscript, such as structure-property relation, or from the point view of energy.

Reply: Thank you for your suggestion. As you suggested, we performed the density functional theory (DFT) calculations and corresponding discussions into the revised manuscript, including the energies and the densities of states (DOS) of three different phases, as shown in Fig. R1 (also Supplementary Fig. 10) to enrich the manuscript from structure-property relation and the point view of energy.

According to the DFT results, the band gaps of γ , β and α phase are 1.76, 1.47 and 1.35 eV, respectively (Fig. R1a-c). At the room temperature, the γ phase has a lower potential energy (-15.45 eV as shown in Fig. R1d), which is consistent with the phase structure we observed at ambient temperature. It is worth noting that the transition structure we discussed in Fig. 3, which differs from the three perovskite phases, is also optimized and shown in Fig. R1e. By calculating the band structure, it is found that a series-connected type II heterojunctions are formed between α -transition phase- γ in Fig. R1f, which can effectively reduce the electron hole pair recombination and improve the photoelectric conversion efficiency. We have added more discussions as follows:

“Then, we used the density functional theory (DFT) calculations to show how these observed phases affect the properties and stabilities of perovskite nanocrystals, as

shown in Supplementary Fig. 10. The densities of states (DOS) of different phases of CsPbI₃ indicate a reduced band gap from the γ to α phase. The lowest potential energy of the γ phase of CsPbI₃ explains why it exists dominantly at the room temperature. The transition structure we discussed in Fig. 3 was also optimized. Based on the band gap calculations, it is worth noting that a series-connected type II heterojunctions are formed between the transition region of α to γ phase, which may effectively reduce the electron-hole pair recombination and improve the photoelectric conversion efficiency. In further study, it will be very interesting to discover corresponding new properties in such transition phase.”

Fig. R1 DFT calculations of CsPbI₃ in different phases and transition structure we discussed in Fig. 3. **a-c**, Calculations of DOS in different phases. The band gaps of γ , β and α phase are 1.76, 1.47 and 1.35 eV, respectively. **d**, Calculations of potential energies of different phases ($E_\gamma < E_\beta < E_\alpha$), which shows that the γ phase is the most stable at low temperatures. **e**, Simulated structure of the transition phase between γ and α phase. **f**, Calculated band gap of transition phase, which is closed to the band gap of β phase.

(2) Structural models in Fig. 1 should be introduced in detail to make it clear, for example which balls represent Cs and Pb and I atoms.

Reply: Thank you for your suggestion. We have marked out these atom balls by using different colors in the structure model in Fig. 1a.

(3) It is not sure that the orthogonal surfaces correspond to the [001] and [110] directions as shown in Fig. 1b, for there many directions perpendicular to the [001], such as [100] and [010].

Reply: Thank you for your comments. The definition of a, b, c axis in the α phase is a little different from those in the β and γ phases. Fig. R2 gives the comparison of the structural model in the α , β , and γ phases and viewed from the [001], [100], [010] and [110] directions. For example, the three orthogonal surfaces of cubic particles in the α phase are (001), (010) and (100) surfaces, while those of cubic particles in the β and γ phases are (001), (110) and (-110) surfaces (please carefully observe Fig. R2). Based on these models in Fig. R2, it is not difficult to identify that the orthogonal surfaces correspond to the [001] and [110] directions instead of [100] and [010]. We have added Fig. R2 into the revised supporting information (Supplementary Fig. 3).

Fig. R2 The structural model in the α , β , and γ phases and viewed from the [001], [100], [010] and [110] directions.

(4) What happen for the figs. 2 c and f, in which there are more noise than other ones.

Reply: Thank you for your comments. Since Fig. 2a-f were obtained under the same acquisition parameters, we think that the more noise in Figs. 2c and f may be caused by different conditions, that is, the thermal effects. On the one hand, thermal vibration can cause atomic blurring in Figs. 2c and f obtained at a temperature of 200 °C, making it difficult to capture an image with higher signal-to-noise ratio. On the other hand, the dynamic rotation of PbI_6 octahedral column during the process of imaging (about 1 min) will also result in the broadening of the intensity peak at the position of ion column, which may be considered as noise in the images. Even in the presence of image blurring in Fig. 2c and f, it will not affect our identification of projected atom position by the profile analysis, since the peak position with high intensity will not be determined by low-intensity noise.

(5) I am confused whether the sample will transform to other phase after heating to 250 C. If so, and all these structures will be compared as shown in Figs. j and k, the data on 250 C are suggested to be shown in the manuscript other than in the extended data.

Reply: Thank you for your comments. At 250 °C, we only find the CsPbI_3 crystals with near α phase. After heating up to 300 °C, as we have already shown in Fig. 4d-g, the CsPbI_3 crystals transformed into Pb and non-perovskite phases Cs_4PbI_6 , the elemental mapping and lattice structures of which are given to confirm this process. In our manuscript, we mainly discussed the transition and dynamics of three perovskite phases (α , β and γ phase) with different structures. However, the non-perovskite phase Cs_4PbI_6 does not satisfy the structural changes in Fig. 1 and 2 caused only by the rotation of PbI_6 octahedrons. That is why we did not merge the results of this section into Fig. 2 and further discussed them in Fig. 4 separately. Meanwhile, we decided to include more plots of the statistical results and conclusions in Fig. 2 instead of placing

too many images causing visual repetition and fatigue. If readers are interested in the results of 150 and 250 °C, they can further review the supplementary data easily.

Reviewer #3:

Thank you for your time and efforts on our manuscript. We are glad to see your recommendation for our work. In your comments, you have raised some professional suggestions about the integrity of the article that can really help us make necessary supplements and improvements to this manuscript. Based on all your comments and suggestions, we have carefully revised the following responses. We sincerely request that you consider our revision for further publication.

(1) One of the main findings of the work is the possibility of mapping the octahedra tilting angles (θ_1 and θ_2) in CsPbI_3 , upon T-variation. First, I would suggest removing from the article the claim that is not feasible to distinguish changes in these angles from X-ray diffraction (any structural paper on Lead halide perovskites from single crystal to powder diffraction, report the two distinct angles for the orthorhombic gamma phase). Second, it is not clear to me how the authors have properly oriented the nanocrystals under the electron beam, to be able to “label” the crystallographic directions that they claim to map ([001] and [110]): in the articles, I cannot find any detail about the strategy adopted by the authors in this sense; how are they sure that they are looking at those specific crystallographic directions? The two/three crystallographic directions in lead halide perovskites are very similar in terms of lattice periodicity, so it's very difficult to distinguish between them.

Moreover, it's not clear to me how the “penetration” studies of the electron beam have been performed. How is it possible to separate the influence from the surface/core in the nanocrystals analyzed? I suppose that the electron beam has a certain penetration depth and what is mapped is an average structure within this portion of nanocrystals analyzed.

Reply: Thank you for your comments. We will reply to your questions one by one as follows:

1. We agree with your comments that diffraction methods can be used to distinguish different phases of perovskites. Here we are going to emphasize that compared with

electron microscopy, which can image the local structure of a single crystal particle, the diffraction methods can provide only the averaged results of samples. Combining these two methods (diffraction and electron microscopy) help us better understand the structural evolution of materials in both the bulk and local structures. As you have suggested, we have modified some of the statements and remove the controversial claim in the introduction paragraphs.

2. During the imaging process, we did not artificially orient crystals to let the right crystallographic axis parallel to the electron beam. This is because that the perovskites are very sensitive to electrons and the alignment of perovskite crystals cause damage to them inevitably (it is an unsolvable technical limitation). However, fortunately, the imaged crystals are cubic or cuboid, so that one of three orthogonal directions of a cubic crystal may align perfectly with the electron beam and the atomic arrangement in this projection can be clearly imaged. We have added these discussions in the Materials and Methods section.

Then, the specific crystallographic directions are determined by comparing the images of these projections directly (experimental images and simulated images in Fig. 1). In Fig. R1 (also Supplementary Fig. 3), we compared the structural models in the α , β , and γ phases viewed from four directions. The three orthogonal surfaces of cubic particles in the α phase are (001), (010) and (100) surfaces, while those of cubic particles in the β and γ phases are (001), (110) and (-110) surfaces (please carefully observe Fig. 1 and Fig. R1). If we can atomically resolve this structure, it is not difficult to find the difference between different projections in different phases by the positions of ion columns (or the rotation angles of PbI_6 octahedrons) as discussed in Fig. 1.

3. The phase distribution was studied in the [110] projection of perovskite crystals rather than the direction along electron beam. The rotation angles in the core or on the surface can be directly measured in the images. Although the 2D projection bring all the information of the column along electron beam, the small rotation angles of PbI_6 octahedrons in the several layers on the surface will not affect overlapped information of the large angles in the deeper bulk area. Therefore, we can observe the different

rotation angles (by the projected atomic positions) in the bulk and on the surface, and it just indicates the phase distribution in this crystal and the phase transition dynamics in the in-situ experiment.

Fig. R1 The structural model in the α , β , and γ phases and viewed from the [001], [100], [010] and [110] directions.

(2) 20 nm nanocrystals cannot be considered Quantum dots (as it is stated in the article). With this nanocrystals size, we are far away from quantum-confinement (being the Bohr radius for these materials something close to 4-5nm), and (on the structural and thus on the point of view of the electronic properties) very close to the bulk material; therefor the classical phase transition scheme from the gamma to the alfa phase reported in this work is not a huge novelty, being already reported for lead halide perovskites bulk materials. The novelty can be the methodology used, but it must be properly described in the manuscript.

Also, the shape of the nanocrystals analyzed is not clearly described (if isotropic makes it really difficult, I would say impossible, to properly orient the different crystallographic directions with respect to the electron beam).

Reply: Thank you for your comments. We respond to each of your requirements one by one as follows:

1. As you suggested, we have changed the term “quantum dots” to “nanocrystals” in the revised manuscript.
2. Compared to the previous study, the imaging methods can provide the changes of local structures during the phase transition with atomic resolution. For example, we found that the rotation angles of PbI_6 octahedrons are different in the bulk and on the surface, that is, there is an obvious distribution of phases at different local structures. However, diffraction methods can only provide periodic and averaged information of phase transition. During the phase transition, on the one hand, we found that the rotation of PbI_6 octahedrons occurs continuously and slowly, and there will be many intermediate angles between the complete γ phase and α phase. The specific rotation angles and rotation speeds are difficult (impossible) to quantitatively describe through diffraction methods, while the atomic imaging method can help resolve them directly in an in-situ experiment. On the other hand, we also found that the speeds of phase transition in different regions of particles (bulk and surface) are completely different, and the phase distribution became more obviously during the heating process, which can only be studied by the atomic imaging method. Our observations confirm that there is a penetration depth (about 5 nm) of thermally induced phase transition on the particle surface and how these phases evolved in a single crystal with temperature and time for the first time. The novelty of this work is definitely not a simple conclusion of the classical phase transition scheme from γ phase to α phase or a stage to show the low-dose imaging techniques. We have systematically quantitatively and intuitively answered how this phase transition process occurs at the atomic scale, how to measure the dynamics of phase transition, how the speed of phase transition changes with time, space, and temperature, and so on. These issues are of significance for understanding the phase stability of perovskites, as well as the structural change and its dynamics during the device preparation, application, and degradation.
3. The shape of the nanocrystals can be easily analyzed by the HAADF-STEM image with low magnification (Fig. R2). Although we can only provide 2D projection of

these particles, it is not difficult to determine that they are cubic (or cuboid) crystals with about 10-30 nm size after observing of a large amount of particles.

Fig. R2 HAADF-STEM images showing cubic perovskite nanocrystals.

(3) CsPbI_3 bulk is not stable at ambient conditions in the perovskite gamma phase but it rapidly transforms to another orthorhombic phase, which is a non-perovskite one (the so-called “yellow” phase). Have the authors considered the possibility of having this phase at room temperature? What would their simulation/intensity profiles look like in that case?

Reply: Thank you for your questions. As you mentioned above, the perovskite phase is unstable at room temperature and will transform into a non-perovskite yellow phase under the influence of water and oxygen in the environment. However, in this work, we don't want to see interference from the yellow phase to the transition between the black phases (from γ phase to α phase). To exclude the interference, we performed the imaging experiment as soon as possible after sample preparation, which was stored in a non-polar n-hexane solvent, to avoid contact with the air environment as much as possible. Therefore, we have not found the yellow phase in our samples at the room temperature. Fig. R3 gives the structural model of the yellow phase, from which we can clearly distinguish the atom positions in the yellow phase from those in the black phases, which confirms that the structures we imaged are all black phases (Supplementary Fig. 3). Then, at over 200 °C, some of nanocrystals will merge into

larger particles, which may be the sources of non-perovskite peaks found in the PXRD results in Supplementary Fig. 8 and previous study¹. However, due to the lack of regular morphology of large particles, we cannot find their crystallographic axes under electron beam, and the element ratios in the perovskite and non-perovskite phases are completely the same. Therefore, we cannot prove whether these large particles are non-perovskite phases only by electron microscopy. On the contrary, no matter at low temperature or high temperature, the perovskite phase particles will maintain a clear crystallographic orientation, allowing us to obtain atomic-resolution images of perovskite structures. These results are also in line with the main idea of our work: revealing the phase transition process and its dynamics of perovskite phases (also black phases) represented by the rotation of PbI_6 octahedrons using atomic-level imaging method.

Fig. R3 The structural models of the yellow phase.

(4) The definition of L1-L4 is not very clear: from what I've understood from Figure 2 the authors have considered a single Cs-I peak separation to compute these lengths. Or several values have been averaged within a single (or even multiple) nanocrystal?

Reply: Thank you for your comments. Just as you understand, L1-L4 are measured in the intensity profiles based on the projected distance between a pair of Cs and I peaks as we defined in Fig. 2, where we give a clear definition in the revised figure. Here we also define L1-L4 in the corresponding projection of model, as shown in Fig. R4 (also in Supplementary Fig. 5), to better understand these four lengths in two different projections, respectively. Due to the rotation of PbI_6 octahedron, they will be different in different phases. For example, when changing from γ phase to α phase with

temperature, L1 gradually increases to approach L2 in the [001] direction, while L3 gradually decreases to approach L4 in the [110] direction. Then, the statistical results of distance ratio (L1/L2 and L3/L4) in Fig. 2j were obtained by the measurements in more pairs of Cs and I peaks (see the average value and error bar in Fig. 2j), which are used to describe the phase transition quantitatively.

Fig. R4 Schematic diagram of the definition of L1-L4 in Fig. 2. The corner marks of L1-L4 (α , β , and γ) indicate the phase type of CsPbI₃.

(5) How many nanocrystals have been analyzed? Do they show the same results in terms of phase transition and octahedra rotations?

Reply: Thank you for your comments. At least ten nanocrystals were studied to give the statistical results in Fig. 2j and k at each temperature with nearly the same phase transition process, indicating the reliability of our conclusions on the phase transition of perovskites.

(6) The use of crystallographic directions throughout the article is done without clearly stating which is the reference system.

Reply: Thank you for your suggestion. The crystallographic directions of the different phases studied in this work are shown in Fig. R1 (also Supplementary Fig. 3).

Reference:

1. S. Wang, *et al.*, Thermal tolerance of perovskite quantum dots dependent on A-site cation and surface ligand. *Nat. Commun.* 2023, 14, 2216.

Reviewer #4:

Thank you for your time on our manuscript. We are glad to see your recommendation for our work. In your comments, you have raised some professional suggestions that can truly help us to make necessary supplement and improvement to this manuscript, and also bring a lot of inspiration to our future work. We have made a careful revision based on the following replies to all your comments and suggestions. We sincerely ask you to consider our revised manuscript for further publication.

(1) how the here found phase transition temperatures compare to previously published values

Reply: Thank you for your comments. In a previous study¹, it was reported that the phase transition points of γ -to- β and β -to- α are at 180 and 280 °C, respectively. In our study, the phase transition was described by the rotation angles of PbI_6 octahedrons indicating the transition points at 150°C and 250°C, respectively. Thus, the observed results are similar to those in the reference.

(2) how the phase transition depends on NC size; several previous studies have discussed the large reduction of phase transition temperatures with decreasing NC size, summarized e.g. in Mater. Adv., 2021, 2, 47. The authors may provide a QD size histogram in the SI and discuss ideally briefly in the main text) how their sizes and findings relate to size-dependent phase transitions in the literature.

Reply: Thank you for your comments. The sizes of nanocrystals are about 10-30 nm, and as you suggested, we summarized the sizes of nanocrystals in Fig. R1 (also Supplementary Fig. 9). In previous studies^{2,3}, it was observed that the phase transition temperature of perovskite will decrease with the reduction of crystal size measured by the X-ray diffraction (Fig. R2). However, for the crystals with 10-30 nm size, the change in phase transition temperature is not significant (within 20 degrees). Thus, it is reasonable to ignore the influence of size effect on phase transition temperature in

our study. As you suggested, we added the histogram of crystal size in the supporting information and discussed the size effect in the revised manuscript as follows:

“Meanwhile, it was also reported that the phase transition temperature decreased with the reduction of nanocrystal size through the XRD in previous studies^{46,47}. For the nanocrystals with 10-30 nm sizes in this work (as shown in Supplementary Fig. 9), the change in phase transition temperature is not significant (within 20 degrees). It is reasonable to ignore the influence of size effect on phase transition temperature in our study.”

Fig. R1 Histogram of the sizes of nanocrystals in our study. It shows the size of the perovskite crystals is within 10-30 nm.

**Phase Transition between Cubic and Tetragonal Phase
at Different Crystal Size**

Fig. R2 Critical temperature versus particle diameter for MAPbBr₃ PQDs. Copyright 2019, American Chemical Society.

(3) how the environment affects the phase transition; there is a large body of literature highlighting the importance of humidity, pressure, or strain due to different thermal expansion of material and substrate; the current work deals with a very specific and rather uncommon case, i.e. single QDs in vacuum, on a TEM substrate; a comment (best via referring to suitable previous literature) should be made how the reader can connect the found phase-transition here with “real-world” applications at ambient conditions in the applications mentioned by the authors in their intro.

Reply: Thank you for your comments. As you mentioned, many different factors in the real world can influence the structural changes in perovskites⁴⁻⁷. During research, we must conduct a study on each possible parameter with other controlled variables. Among them, the thermal-induced phase transition and structural degradation should be the most direct influencing factors in device applications. Therefore, in our study, we mainly focused on the transition behaviors of perovskite phases with temperature (single parameter) and provided an in-situ experiment and analysis method to identify and investigate this process, although other environments are still a little different from real-world applications, which may create a bias between the results of different methods. Even though in situ heating with atomic imaging in electron microscopy is relatively simplest in-situ experiment, there are still many experimental details that need to be addressed, which makes it difficult for other reported results to obtain a new understanding at atomic-scale as we did. In following research, it is possible to combine high-resolution imaging techniques with in-situ gas and liquid cells to obtain the real-space imaging results on other factors, such as water and oxygen. This is an important first step for in-situ characterization of perovskites achieved by our efforts in this work. Meanwhile, as you suggested, we provided the following comment with references in the revised introduction paragraphs to demonstrate the multiple factors

affecting phase transition and the significance of thermal-induced phase transition we studied here.

“Many different factors in the real world can influence the structural changes in perovskites²⁷⁻³⁰. Among them, the thermal-induced phase transition and structural degradation should be the most direct influencing factors in device applications.”

(4) how to reconcile the slow phase transition rate found here (proceeding over several minutes) to the sub-nanosecond photoinduced phase transitions reported previously (see for the very related CsPbBr₃ QDs e.g. Nature Communications 2019, 10, 504, or for MAPbI₃ bulk e.g. Matter 2023, 6, 2, 460).

Reply: Thank you for your comments. In these references, CsPbBr₃ QDs experienced significant impulsive heating following moderate-to-high-fluence photo excitation, and then lattice undergoes a sub-nanosecond photoinduced phase transitions revealed by time-resolved X-ray diffraction. Although this reference and our work are all the thermal-induced phase transition processes, the way how the sample excited by heat is totally different. The heating experiment in our study was conducted using an in-situ heating chips (Protochips, Fusion 350), which is a heating method that changes the ambient temperature and allows heat to gradually penetrate from the surface to the center of particles. In the references, the heat was applied by pulsed laser and rapidly excites the simultaneous phase transition on the surface and inside. The difference between these two processes is just the huge speed difference in heat transfer from the surface to the center, which is the dynamic behavior of phase transfer (or heat transfer) that we are most concerned about.

(5) The discussed directions/projections seem to correspond to the pseudocubic system. This is important and, hence, should be mentioned and briefly discussed to

- clarify that those do not refer to the respective tetragonal and orthorhombic systems (which are different!)

- discuss the peculiar situation that the shown NCs seem to exhibit a surface termination along the pseudocubic axes, irrespective of the fact that they are orthorhombic. Is this representative for the studied NC batch? And how does this relate to other previous studies of CsPbI₃ NCs?

Reply: Thank you for your comments. The CsPbI₃ nanocrystals studied in this work are basically cubic or cuboid particles representative for the studied crystal batch. The three orthogonal directions of cubic particles correspond to the [001], [110], and [-110] directions of the β and γ phases or the [001], [010], and [100] directions of the α phase, which are a little different due to their different crystallographic systems (cubic for α , tetragonal for β , and orthorhombic for γ). As shown in Fig. R3, the commonly defined crystallographic projections of different phases are given. Based on the images in Fig. 1 and 2, what we observe are basically very regular cubic (or cuboid) crystals, which indicates that they are indeed in cubic, tetragonal, or orthorhombic systems rather than the pseudocubic system. Meanwhile, since we identify different phases and their distribution mainly by measuring the rotation angle of each PbI₆ octahedron column, the overall particle morphology and axis deviation caused by such rotation may not be important, which is not the core phenomenon and viewpoint discussed in this work.

Fig. R3 The structural model in the α , β , and γ phases and viewed from the [001], [100], [010] and [110] directions.

(6) The experimental methods section in the SI is rather short. However, some more info would be useful for the reader, e.g. the precise vacuum and substrate employed.

Reply: Thank you for your comments. We have added more details of parameters in imaging experiment into the Materials and Methods section.

(7) The labelling of figures and captions is not always complete and sometimes even inconsistent and confusing.

- Fig. 1, 2, and 3: More details should be given on how exactly the intensity profiles were obtained. Is some averaging perpendicular to the profile direction involved? This info may be given in the SI section, while the main text/caption can then refer to this section.

Reply: Thank you for your comments. All the intensity profiles were extracted with only one-pixel width, and the intensity of each point just represents the intensity of this pixel. Therefore, there is no averaging involved perpendicular to profile direction. As you suggested, we have added it into the Materials and Methods section.

(8) Fig. 1c, d, f, g: the precise region from which (d) was extracted should be indicated in (c) with a box; same for (f) and (g). This is important as the surface-near QD regions behave differently than the QD interior, as shown later in the manuscript.

Reply: Thank you for your comments. The regions of Fig. 1d and 1g are located in the bulk of their crystals and have been outlined by different colored boxes.

(9) Regarding inconsistent (and therefore confusing) color choice in Fig. 1:

- Fig. 1a: red and blue distinguish angles θ_1 and θ_2 , respectively
- Fig. 1d and 1g: red and blue indicate some line profile along different directions than those involved in defining the angles θ

- Fig.1e and 1h: red and blue distinguish γ and beta phase, respectively

These inconsistencies should be avoided, and the color choice improved, to better guide the reader.

Reply: Thank you for your suggestion. As you suggested, we have changed the color that we used in Fig. 1.

(10) Fig. 3: the labelling of subfigures and captions for subfigures (d) and (f) is confusing and could be improved. Furthermore, essential info (e.g. on timing within the temperature ramp) is omitted. Few suggestions:

- do not use Fig. no. 1, 2, 3 (for the sub-sub-figures), as this is easily confused with Fig. 1, 2, 3 (reserved for the main figures). Use instead for example labels (i), (ii), and (iii). This means each panel can be unambiguously identified, as Fig.3d(i), etc.

- The new labelling should be applied to both images in (d) and the respective axis in (f)

- Better still would be if the consecutive "index" labelling 1-6 would be replaced with actual time stamps (I presumed it should be 1.5 min, 3 min, 8 min, 9.5 min, 11 min, 15 min?) along with info on when (at 8 min?) the temperature was changed from 100 to 200 deg C. This would mean adding labels to the images in (d) and changing the labelling of the "time" axis in (f). All info in (d) and (f) should allow the reader to unambiguously connect to the data points shown in (g).

- Fig. 3 g,h: it should be specified which rotation angles is given, i.e. θ_1 or θ_2 .

Reply: Thank you for your comments. As you suggested, we changed "Fig. no. 1-6" in Fig. 3 and Extended Data Fig. 4 to the label "i-vi". Images in Fig. 3d are extracted from the same area in different images (figures i-vi in Extended Data Fig. 4) of phase transition regions at the subsurface.

Then, according to the shooting time of each image in Supplementary Fig. 7, we also changed "index" labelling 1-6 to the actual time stamps (1.5 min, 3 min, 4.5 min, 9.5 min, 11 min, 12.5 min as you noticed).

As we defined θ_1 and θ_2 in Fig. 1a, the projection in Fig. 3 is viewed from the [001] direction, and the measured angle is corresponding to θ_1 . It is specified in the legends of all these figures.

References:

1. A. Marronnier, *et al.*, Anharmonicity and disorder in the black phases of cesium lead iodide used for stable inorganic perovskite solar cell. *ACS Nano* 2018, 12, 3477-3486.
2. A. Alaei, *et al.*, Polymorphism in metal halide perovskites. *Mater. Adv.* 2021, 2, 47-63.
3. L. Liu, *et al.*, Size-dependent phase transition in perovskite nanocrystals. *J. Phys. Chem. Lett.* 2019, 10, 5451-5457.
4. J. A. Steele, *et al.*, Thermal unequilibrium of strained black CsPbI₃ thin films. *Science* 2019, 365, 679-684.
5. S. Chen, *et al.*, General decomposition pathway of organic-inorganic hybrid perovskites through an intermediate superstructure and its suppression mechanism. *Adv. Mater.* 2020, 32, 2001107.
6. B. Turedi, *et al.*, Water-induced dimensionality reduction in metal-halide perovskites. *J. Phys. Chem. C* 2018, 122, 14128-14134.
7. J. He, *et al.*, Why oxygen increases carrier lifetimes but accelerates degradation of CH₃NH₃PbI₃ under light irradiation: time-domain ab initio analysis. *J. Am. Chem. Soc.* 2020, 142, 14664-14673.

REVIEWER COMMENTS

Reviewer #1 (Remarks to the Author):

I recommend the revised version of this manuscript for publication.

Reviewer #2 (Remarks to the Author):

The authors have answered all my questions. I think it can be accepted in this version after one point is addressed that some changes have been made on the author lists. Please give some explanations.

Reviewer #3 (Remarks to the Author):

I've read the revised version of the manuscript, along with the authors' replies and I still have some concerns about the methodology adopted.

The authors report the structural models (e.g. in Figure 1) of the three known lead halide perovskite phases, as well as simulations computed from these models, and then compare these simulations with experimental data.

Where do these models come from? Are they from the literature? As I mentioned in the previous reviewer's report, the structure- unit cell orientation (at least in the gamma phase) depends on the crystallographic reference system. If we don't know which is the reference system, the crystallographic direction [100], [001], etc does not tell us anything about the structure of the material.

Moreover the distances between atoms can change from sample to sample, for example they change by moving from the bulk to the nanoscale: in figure 2e,h the distances between atomic columns computed from the "reference structure" are compared with the experimental data and used to identify the crystalline phase.

But these reference data taken from the literature (I suppose) have not been optimized against the experimental data, so the offset that the authors highlight (e.g. in Figure 2h the beta phase intensity peaks are systematically shifted towards lower distances) may be simply a consequence of this usual structural variability in terms of lattice periodicity (and therefore of interatomic distances) that is very common by moving from bulk to different sizes of lead halide perovskite nanocrystals.

Reviewer #4 (Remarks to the Author):

Boyuan Shen et al. have significantly revised their original manuscript. Since all my initial concerns have been adequately addressed, I can now suggest publication in Nature Communications without further revision.

RESPONSE TO REVIEWERS' COMMENTS

Reviewer #1:

Thanks very much for your kind work and consideration on publication of our paper.

On behalf of my co-authors, we would like to express our great appreciation to you.

Reviewer #2:

Thank you very much for your time on our work. On behalf of my co-authors, we would like to express our great appreciation to you.

During the preparation of this manuscript, M. Ma and Prof. B. Shen analyzed the imaging data and wrote the manuscript together. Then, according to the comments in the first-round review, M. Ma added relevant experiments and completed the revision of this manuscript under the guidance of Prof. B. Shen. After all these considerations, we adjust M. Ma to be the first author (first contributor to this work), and Prof. B. Y. Shen as the corresponding author at the end of the author list (academic leader of this work). Meanwhile, we add two authors to the author list, X. Liang and Prof. C. Tao, who are responsible for the DFT energy calculations shown as Supplementary Fig. 10 in the revised manuscript. We have updated the contents in the Author Contributions section.

Reviewer #3:

Thank you for your time on our manuscript. In your current comments, we understand that you are still concerned about the origin the models used in our analysis, and the differences between them, so we gave more information (highlighted in the revised manuscript) about the phase identification of perovskites. Based on all your comments and suggestions, we have carefully revised the manuscript and provided following replies. We hope these replies can address all your concerns on our research, and we sincerely ask you to reconsider our revised manuscript for further publication.

(1) The authors report the structural models (e.g. in Figure 1) of the three known lead halide perovskite phases, as well as simulations computed from these models, and then compare these simulations with experimental data.

Where do these models come from? Are they from the literature? As I mentioned in the previous reviewer's report, the structure- unit cell orientation (at least in the gamma phase) depends on the crystallographic reference system. If we don't know which is the reference system, the crystallographic direction [100], [001], etc does not tell us anything about the structure of the material.

Reply: Thank you for your questions. First, the structural models of the three phases of perovskites introduced in our manuscript were obtained from Reference [1] using the synchrotron X-ray powder diffraction (SXRD). Then, as you suggested, we have added the crystallographic reference systems in Fig. 1a (also Fig. R1). In fact, the crystal structures and phases of different perovskites at different temperatures (e.g., CsSnI₃, CsPbCl₃ and CH₃NH₃PbX₃, X=Cl, Br, I) were experimentally measured and analyzed by the Rietveld refinement of PXRD method and neutron-scattering in about the 1970- to-90s.²⁻⁴ Nowadays, over 30 years later, all its crystallographic phases have reached a comprehensive understanding (experimental and theoretical).⁵ Based on the previous studies,^{2,6} we showed the coordinate system diagram of different phases in Fig. R2, which directly illustrates the relationship between them. In Fig. R2a and b, we can see that the unit cells of the β and γ phases have the same orientation relationship, different

from that of the α phase. Therefore, the directions of the new principal axes of the β and γ phase lattice, $[100]_{\beta,\gamma}$, $[010]_{\beta,\gamma}$ and $[001]_{\beta,\gamma}$ (subscripts “ β ” and “ γ ” represent β and γ phase crystal structure, respectively), are taken along $[110]$, $[1\bar{1}0]$, and $[001]$ referred to the pseudocubic lattice, respectively.

Fig. R1 Identifying the phase structures of CsPbI_3 nanocrystals.

Fig. R2 The difference of coordinate systems between the cubic phase perovskites and other phases. **a.** Relation between the reciprocal lattice of the tetragonal structure drawn with the thick lines and that of the cubic structure drawn with the thin lines. Copyright 1974, American Physical Society. **b.** Geometrical relationship between cubic (red dash lines) and orthorhombic (blue dash lines) unit-cell axes and faces. Copyright 2022, American Chemical Society.

(2) Moreover the distances between atoms can change from sample to sample, for example they change by moving from the bulk to the nanoscale: in figure 2e,h the distances between atomic columns computed from the "reference structure" are

compared with the experimental data and used to identify the crystalline phase.

But these reference data taken from the literature (I suppose) have not been optimized against the experimental data, so the offset that the authors highlight (e.g. in Figure 2h the beta phase intensity peaks are systematically shifted towards lower distances) may be simply a consequence of this usual structural variability in terms of lattice periodicity (and therefore of interatomic distances) that is very common by moving from bulk to different sizes of lead halide perovskite nanocrystals.

Reply: Thank you for your questions. Based on your descriptions, we assumed that you are wondering why the distance between the atomic columns can identify crystalline phases in Fig. 1e and 1h, rather than in Fig. 2e and 2h. First, in Fig. R3b and 3d, we supplemented the structural models of different phases along the arrows in Fig. R3a and 3c, respectively, to link the changes in profile peak positions with the changes in atomic positions in the models. In Fig. R3b, we clearly see the change in the distance between I columns (marked by black dash lines) in the β and γ phases, which is different from those in the α phase caused by the rotation of PbI_6 octahedrons. In Fig. R3d, we can see that the Cs columns of γ phase deviate from the center of the surrounding four Pb-I columns, which is different from the strictly centered Cs ions distribution in the β and α phases. These obvious structural features are not only directly visible in the structural model, but can also be identified in the intensity profiles. It is worth noting that, in Fig. R3a and 3c (also Fig. 1e and 1h), the solid lines were extracted from the images, and the dash lines were extracted from the simulated images based on the structural models. Compared the experimental and simulated images, it is clear that two intensity profiles are consistent with the structural features of γ phase. At room temperature, the structural features (the shifts of peaks in these profiles) come from the periodic information of bulk lattices (the images and profiles are extracted from the center of particles as shown in Fig. 1), of course not due to differences in the samples.

More importantly, such method for identifying perovskite phases was not used for the first time by us, but rather a well-recognized method to use these structural features of γ phase, including our previous work (Adv. Mater. 2023, 35, 202300653). For example,

as shown in Fig. R4 (obtained from Reference [7,8]), these structural features were used to confirm the γ phase of CsPbI₃ nanocrystal. In Fig. R4a, just viewed from the [001] direction, the rotation of PbI₆ octahedrons can also be observed as we have imaged. In Fig. R4b and 4c, the FA ions were doped to change the phase of CsPbI₃ and the samples were viewed from the [110] direction. At a low FA doping degree (Fig. R4c left), the same feature (the off-center A site ions) was observed as we have imaged. At a high FA doping degree (Fig. R4c right), such feature disappeared to indicate a phase transition induced by FA doping.

Moreover, the γ phase of CsPbI₃ nanocrystal and its transition were not only confirmed by the images, but also supported by the PXRD results in Supporting Information (we have added during the last-round revision). Therefore, both the process (a widely used method) and results (also supported by other methods) of our phase identification are correct and independent of the difference in peak shifts between samples.

Fig. R3 Phase identification of perovskites. **a.** IDPC-STEM image, the simulated image and extracted simulated profiles of α , β and γ CsPbI₃ viewed from the [001] direction. **b.** Corresponding structural models of α , β and γ CsPbI₃. **c.** IDPC-STEM image, the simulated image and extracted simulated profiles of α , β and γ CsPbI₃ viewed from the [110] direction. **d.** Corresponding structural models of α , β and γ CsPbI₃.

Fig. R4 Atomic arrangement characteristics of different phases. a. HAADF images of the of γ phase CsPbI_3 in $[001]$ direction. The PbI_6 octahedrons rotated, and the overlapped Cs columns transformed into oval shape, as marked by blue ellipses. Copyright 2018, American Chemical Society. **b.** HAADF image of an orthorhombic $\text{FA}_{0.15}\text{Cs}_{0.85}\text{PbI}_3$ grain in $[110]$ direction. **c.** Fine atomic structures and corresponding intensity profiles along the orange arrows, indicating the off-center A sites in the left part, but the centered A sites in the right part. Copyright 2022, American Chemical Society.

References:

1. Marronnier, A.; *et al.* Anharmonicity and disorder in the black phases of cesium lead iodide used for stable inorganic perovskite solar cells. *ACS Nano* 2018, **12**, 3477-3486.
2. Fujii, Y.; Hoshino, S.; Yamada, Y.; Shirane, G. Neutron-scattering study on phase transitions of CsPbCl_3 . *Phys. Rev. B* 1974, **9**, 4549.
3. Poglitsch, A.; Weber, D. Dynamic disorder in methylammoniumtrihalogenoplumbates (II) observed by millimeter-wave spectroscopy. *J. Chem. Phys.* 1987, **87**, 6373-6378.

4. Yamada, K.; Funabiki, S.; Horimoto, H.; Matsui, T.; Okuda, T.; Ichiba, S. Structural phase transitions of the polymorphs of CsSnI₃ by means of Rietveld analysis of the X-Ray diffraction. *Chem. Lett.* 1991, **20**, 801-804.
5. Tan, W. L.; McNeill, C. R. X-ray diffraction of photovoltaic perovskites: Principles and applications. *Appl. Phys. Rev.* 2022, **9**, 021310.
6. Liu, J. K.; *et al.* Self-assembly and regrowth of metal halide perovskite nanocrystals for optoelectronic applications. *Acc. Chem. Res.* 2022, **55**, 262-274.
7. Sun, J. K.; *et al.* Polar solvent induced lattice distortion of cubic CsPbI₃ nanocubes and hierarchical self-assembly into orthorhombic single-crystalline nanowires. *J. Am. Chem. Soc.* 2018, **140**, 11705-11715.
8. Cai, S. H.; *et al.* Atomically resolved electrically active intragrain interfaces in perovskite semiconductors. *J. Am. Chem. Soc.* 2022, **144**, 1910-1920.

Reviewer #4:

Thank you very much for your time on our work. On behalf of my co-authors, we would like to express our great appreciation to you.

REVIEWERS' COMMENTS

Reviewer #3 (Remarks to the Author):

The manuscript can be accepted in its present form, all criticisms have been addressed

RESPONSE TO REVIEWERS' COMMENTS

Reviewer #3:

We are very glad that our responses answered all your concerns. Thanks very much for your kind work and consideration on publication of our paper. On behalf of my co-authors, we would like to express our great appreciation to you.